# TPU-GAN: LEARNING TEMPORAL COHERENCE FROM DYNAMIC POINT CLOUD SEQUENCES

**Zijie Li**
Department of Mechanical Engineering
Carnegie Mellon University

**Tianqin Li**
School of Computer Science
Carnegie Mellon University

**Amir Barati Farimani**
Department of Mechanical Engineering
Carnegie Mellon University

## ABSTRACT

Point cloud sequence is an important data representation that provides flexible shape and motion information. Prior work demonstrates that incorporating scene flow information into loss can make model learn temporally coherent feature spaces. However, it is prohibitively expensive to acquire point correspondence information across frames in real-world environments. In this work, we propose a super-resolution generative adversarial network (GAN) for dynamic point cloud sequences without requiring point correspondence annotation. Our model, Temporal Point cloud Upsampling GAN (TPU-GAN), can implicitly learn the underlying temporal coherence from point cloud sequence, which in turn guides the generator to produce temporally coherent output. In addition, we propose a learnable masking module to adapt upsampling ratio according to the point distribution. We conduct extensive experiments on point cloud sequences from two different domains: particles in the fluid dynamical system and human action scanned data. The quantitative and qualitative evaluation demonstrates the effectiveness of our method on upsampling task as well as learning temporal coherence from irregular point cloud sequences.

## 1 INTRODUCTION

Temporally evolving point clouds are ubiquitous in various fields, e.g. particles in the $N$-body system, or real-world moving objects captured by LiDAR sensor or RGB-D camera. Understanding the spatiotemporal pattern lies in these sequential data are of great importance to engineering and scientific disciplines. However, high-resolution point cloud sequences can be very expensive to obtain. In practice, high-resolution physics simulation demands substantial computational resources and 3D scanned data are often sparse and noisy due to hardware constraints. Impressive progress has been made by applying learning-based algorithms (Yu et al., 2018b; Li et al., 2019; Wu et al., 2019; Yifan et al., 2019; Qian et al., 2021) to reconstruct a dense and clean point cloud given the observed sparse ones. While these works provide powerful framework for upsampling static point clouds, the change of points with respect to time is not taken into consideration. Prantl et al. (2020) demonstrates that supervising the scene flow (displacement of points) of generated points during the training process can greatly stabilize the learned latent space for point cloud super-resolution task. Yet this requires knowing the point correspondence across the frames to derive the scene flow , which is expensive and sometimes impossible to acquire in real-world scenarios. For 3D scanned data, the point clouds in two frames do not necessarily have correspondences and the number of points may change.

In this case, generative adversarial networks (GANs) (Goodfellow et al., 2014) provides an attractive alternative for learning useful generative and discriminative representation without label and heuristic loss function (e.g. L2 loss that penalizes the scene flow difference). Under the context of point cloud sequence's super-resolution, we can use the discriminator to learn both the spatial and temporal patterns from data implicitly. With these learned spatiotemporal discriminative representation, we can further use them to guide the generator to produce output that are spatially and temporally coherent.

More formally, in this work we present a GAN framework for super-resolution on the dynamic point cloud sequences. By experimenting on point cloud sequences from two very different domains - fluid particles and 3D scanned human action, we show that the proposed upsampling model can produce output that are spatially and temporally consistent with target data without explicit supervision on the point displacements across frames. We also show that the discriminative representation learned by temporal discriminator can be used for classification task and provides competitive performance. Furthermore, we propose a learnable masking module to adaptively upsample points that have a irregular density distribution. This allows model to produce non-uniform point distribution that matches the target domain.

## 2    RELATED WORKS

**Deep learning on static point clouds**    The pioneering work PointNet (Qi et al., 2017a) carries the success of deep neural networks on 2D structured grid data to the field of unstructured 3D point cloud processing. PointNet proposes a neural network architecture with multi-layer perceptron (MLP) shared acrossed points and aggregation mechanism that respects permutation invariance in the input. Following PointNet, a wide array of works had contributed to improve and extend the neural representation learning paradigm on point cloud data (Qi et al., 2017b; Li et al., 2018; Wang et al., 2019b; Wu et al., 2020; Zhao et al., 2019; Shen et al., 2018). These methods have shown to learn good representation for static point cloud analysis task such as classification, semantic segmentation and object detection.

**Learning-based point cloud upsampling**    With the success in learning-based image super-resolution (Dong et al., 2015; Ledig et al., 2017), a number of works transfer the learning-based super-resolution concept to the 3D point cloud. PU-Net (Yu et al., 2018b) proposes the first learning-based method that upsamples point cloud features with multi-level expansion units. Yu et al. (2018a) extends and improves PU-Net by adding an edge-aware loss to consolidate the upsampled point clouds. Yifan et al. (2019) proposes a progressive point cloud upsampling network which supervises the upsampled point clouds at different levels. Li et al. (2019); Wu et al. (2019) leverage GANs to learn high-quality upsampling result, which demonstrate adversarial training can benefit point cloud upsampling task. Very recent work Qian et al. (2021) proposes Inception DenseGCN and NodeShuffle, which significantly improves the upsampling quality. Another concurrent work PUGeo-Net (Qian et al., 2020) proposes a geometric-centric deep learning framework for learning the local fundamental forms.

**Spatiotemporal representation learning on point clouds**    Different deep learning methodologies for learning discriminative spatiotemporal representation for video classification have been proposed (Ng et al., 2015; Donahue et al., 2016; Feichtenhofer et al., 2016; Carreira & Zisserman, 2018; Tran et al., 2015), and been further explored on the generative tasks including video generation (Tulyakov et al., 2017; Clark et al., 2019; Saito et al., 2020) and super-resolution (Xie et al., 2018; Wang et al., 2019a; Haris et al., 2019; Chu et al., 2020). Following previous approaches that treat video data as volumetric data, Choy et al. (2019); Luo et al. (2020) transform point cloud sequences into volumetric voxel-based representation and apply convolution neural networks on it. Liu et al. (2019b; 2020) extends the set convolution from PointNet++ (Qi et al., 2017b) by establishing neighborhood across different frames. PSTNet (Fan et al., 2021b) proposes a spatiotemporal decoupled point convolution for point cloud sequence processing. Recent works Fan et al. (2021a); Wang et al. (2021a) introduce attention mechanism to better capture data correlation across frames. In terms of generative tasks, Tranquil Clouds (Prantl et al., 2020) proposes a temporal loss that supervises the scene flow across frames and stabilize the generation of point clouds. Rempe et al. (2020) proposes CaSPR, a deep learning framework for learning continuous object-centric spatiotemporal representation from point clouds. Lu et al. (2021) proposes interpolation network to enhance the temporal resolution of point cloud sequence. Another recent work Wang et al. (2021b) proposes a learning-based recurrent temporal alignment module for assimilating temporal information during upsampling sequential point clouds.

In general, our work is closely related to PointNet++ (Qi et al., 2017b) and MeteorNet (Liu et al., 2020) which use set abstraction with different grouping strategies to learn discriminative representation on point clouds. In the generator, we adopt a similar multi-scale feature extractor architecture as the one proposed in PU-GCN (Qian et al., 2021). Compared to PU-GAN (Li et al., 2019), our method takes the change of points in time dimension into account, and uses hierarchical structures in the discriminator instead of operating on full resolution point clouds. Under the hood of sequential point cloud data generative modelling, Tranquil Clouds (Prantl et al., 2020), CaSPR (Rempe et al.,

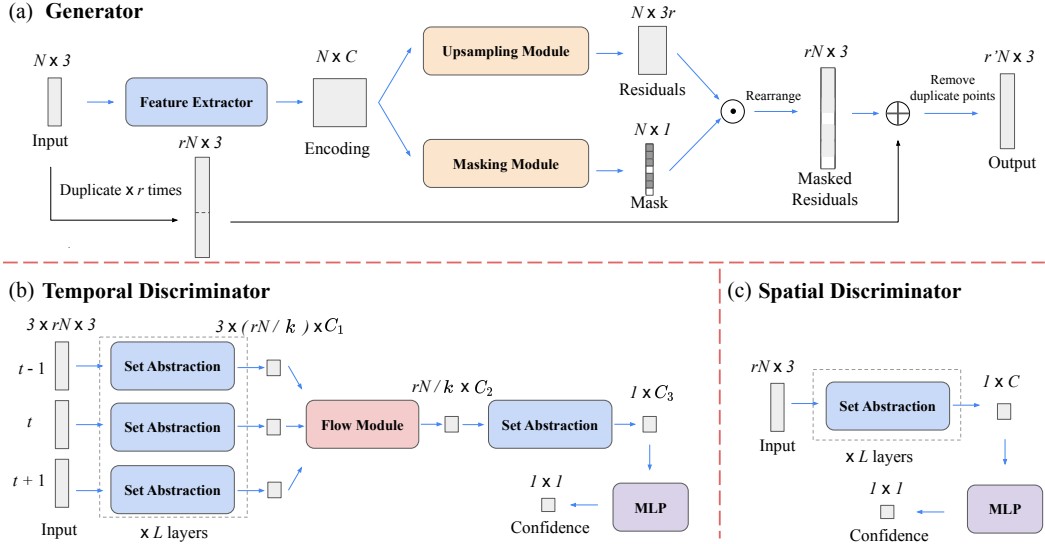

Figure 1: Overview of TPU-GAN's architecture, it comprises a generator, a temporal discriminator and a spatial discriminator. **(a)** The *Generator* takes a coarse point cloud as input and predicts a refined point cloud. Inside the generator, the input is first project into a high-dimensional latent encoding, the encoding is then transformed into residuals by *Upsampling Module* and a binary mask by *Masking Module*. Through an element-wise multiplication ⊙, residuals of unnecessary upsampling will be masked to zeros. **(b)** The *Temporal Discriminator* takes three consecutive frames of point clouds as input, extracts features via set abstraction layesr and flow embedding layers, and predicts the confidence if it is temporally coherent. **(c)** The *Spatial Discriminator* takes a point cloud as input, extracts features via set abstraction layers, and outputs the confidence if it is spatially coherent. (Detailed schematic of each sub-module can be found in Appendix A.1)

2020), and SPU (Wang et al., 2021b) share similar target with our work. CaSPR focuses on object-centric tasks and thus is not available for processing multi-object collections or scenes. Tranquil Clouds requires point correspondence across frames to evaluate the first-order and second-order finite difference loss, which our method does not require during the training phase. Compared to SPU (Wang et al., 2021b), we avoid processing the high resolution outputs multiple times by using temporal discriminator instead of recurrent architectures to learn the temporal coherence.

## 3 METHODS

### 3.1 HIERARCHICAL SPATIAL FEATURE LEARNING

**Set abstraction** Point clouds are irregular and unordered data where traditional convolution operations usually do not fit. To cope with this irregularity, we adopt the set abstraction layer from Qi et al. (2017a;b) to extract features from point clouds for discriminative models in our proposed framework. Inside a set abstraction layer, point cloud is divided into different groups, and then a universal set function approximator is applied to extract the local features of each group

$$F'_i = \gamma \left( \max_{j \in \mathcal{N}(i)} h(F_j, \mathbf{r}_j - \mathbf{r}_i) \right), \tag{1}$$

where $F$ denotes features of points, $\mathbf{r}$ denotes the positions of points, $\mathcal{N}(i)$ denotes the neighborhood of point $i$, $\gamma$ and $h$ are learnable functions which are usually implemented as multi-layer perceptrons (MLPs).

**Multi-scale graph convolution networks** Recent work PU-GCN (Qian et al., 2021) proposes a GCN-based architecture (Inception DenseGCN and NodeShuffle) and demonstrates the superiority of using GCN on the point cloud upsampling tasks. Similar to the set abstraction layer from PointNet, the GCN layer in PU-GCN divides the point cloud into different groups and applies learnable functions to

these groups to extract local features. The major differences are that first, the GCN layer in PU-GCN constructs the neighborhood following DGCNN's (Wang et al., 2019b) formulation by dynamically querying $k$-nearest neighbors in the latent space instead of Euclidean space, and second, in PU-GCN, each layer aggregates features from different scales of neighborhood at each layer. These differences make the Inception DenseGCN layer in PU-GCN more effective at extracting features from different scales and improve the overall performance of model. However, since it is more computationally intensive than the standard set abstraction, we only apply similar GCN layers to the generator, where it is most effective.

## 3.2 TEMPORAL FEATURE LEARNING

To capture the underlying patterns of point cloud sequences over the time dimension, we construct a module that operates on the temporal neighborhood. We adapt the flow embedding layer from Liu et al. (2019b; 2020) to operate on neighborhood across frames. Given two consecutive frames of point clouds $\{X^t, X^{t+1}\}$ and their corresponding features $\{F^t, F^{t+1}\}$, we use the flow embedding layer to mix points across frames and extract temporal features. The mixture of features $\{F^t, F^{t+1}\}$ across two consecutive frames via flow embedding layer is defined as:

$$\tilde{F}_j^t = \gamma \left( \max_{j \in \mathcal{N}(i)} h(F_i^t, F_j^{t+1}, \mathbf{r}_j^{t+1} - \mathbf{r}_i^t) \right), \tag{2}$$

where $j$ denotes the index of points from frame $t + 1$ and $i$ denotes the index of points from frame $t$.

## 3.3 ENFORCING TEMPORAL COHERENCE VIA TEMPORAL DISCRIMINATOR

When the target data are point cloud sequences that come from physical processes like particle dynamics or camera-scanned human action, these data are highly correlated in the time dimension. One way to capture this temporal coherence would be directly using the scene flow information in the ground truth data (i.e. displacements of points) to supervise the training process, i.e. adding an additional loss term that evaluates the scene flow difference between ground truth and prediction. However, for many real-world scenarios, there are often no clear correspondences between point clouds from two frames. Therefore, the displacement of points between different frames are not annotated and cannot be used for training. To enforce the temporal coherence without manually encoding scene flow information, we propose a discriminator to learn the temporal pattern from ground truth data implicitly and use the discriminator to guide generator predicting temporally coherent output.

Our proposed temporal discriminator is built upon set abstraction layer and flow embedding layer discussed in the previous section. The temporal discriminator takes three consecutive frames of point clouds as input. These point clouds are first downsampled and transformed by several layers of set abstraction. Then we apply the flow module, which consists of two flow embedding layers, to assimilate temporal information from the point cloud sequences. Finally, we use set abstraction layer and a MLP to aggregate global information and predict whether the input sequence is valid (temporally coherent) or not.

## 3.4 ADAPTIVE UPSAMPLING

In most of the learning-based point cloud upsampling methods, all the points in the input will be upsampled and the ratio between the number of input points and output points is fixed. Given an upsampling network with a upsampling ratio of $r$, for the point $x_i$ in the input point cloud, there will be $r$ points in the upsampled point cloud that is directly upsampled from the input point $x_i$. Since these networks are trained and operates on local patches, the locality will drive the upsampled points to be very close to the original input point. Every input point will have $r$ neighbors very close to them in the upsampled point cloud. However, not all point clouds are distributed in a uniform way such that every point has similar amount of points in their neighborhood. For instance, in particle-based fluid dynamics, near-surface or flying away particles will have very different density compared to particles in the center of fluid field.

To suppress unnecessary upsampling, we propose a learnable masking module, which predicts a mask $M$ that tells the network if it needs to upsample a point. Concretely, for each point in the input point cloud, the masking module will predict a confidence value $p_i$, which forms the soft mask vector

$M_p = [p_1, p_2, \ldots, p_N]^T$. The soft mask is then quantized and transformed into a binary mask $M_b$ based on predefined threshold $\epsilon$

$$[M_b]_i = \begin{cases} 1, & [M_p]_i \geq \epsilon \\ 0, & [M_p]_i < \epsilon \end{cases}. \tag{3}$$

In practice we set this threshold value to a small number (e.g. $\epsilon = 0.01$[1]) so that the upsampling of a point will be suppressed only when the confidence level is low. Finally, we use the binary mask to modulate the predicted residuals by doing an element-wise multiplication, so that the residuals of unnecessary upsampling will be masked to zero. Note that here we use the binary mask to modulate the output residuals instead of the input encoding of the upsampling module. The reason is that if we mask the features of some points to zero, then the upsampling module cannot aggregate features from these points when predicting the residuals.

During training, we generate the label $M_t$ for masking module by counting the number of neighbors of input points in the target point cloud $Q$

$$[M_t]_i = \begin{cases} 1, & |\mathcal{N}(i)| \geq 3 \\ 0, & |\mathcal{N}(i)| < 3 \end{cases}^{[2]}, \tag{4}$$

where the neighbor set: $\mathcal{N}(i) = \{j | \, ||\mathbf{r}_j - \mathbf{r}_i||_2 < r_0, j \in Q\}$ is derived by querying the neighbor of point $i$ in the target point cloud $Q$. The search radius $r_0$ is set to be slightly larger than the average nearest distance between points in the target point cloud. Based on the generated labels, we use L1 norm to evaluate the loss between soft mask $M_p$ and ground truth $M_t$:

$$\mathcal{L}_{\text{mask}} = \frac{1}{N} \sum_{i=1}^{N} ||[M_p]_i - [M_t]_i||_1. \tag{5}$$

## 3.5 TRAINING

**Model architectures** We follow the design of PU-GCN to build the feature extractor in the generator, where we first Inception DenseGCN to extract features from the input points. Inside the upsampling and the masking module, we use two layers of GCN to aggregate features from neighborhood and use MLP to project the latent encoding into prediction. As for discriminators, we adopt the hierarchical structure similar to the PointNet++, where we use set abstraction layer/flow embedding layer to extract features at different scales and continue to downsample the point cloud through network layers. (Model details can be found in Appendix A.1)

**Loss function** We adopt the least-squared GAN (Mao et al., 2017) setting to train our model. Based on LS-GAN's formulation, we can derive the adversarial loss for temporal discriminator $D_t$ and spatial discriminator $D_s$,

$$\mathcal{L}_{D_t} = [1 - D_t(\mathbb{Y}_t)]^2 + D_t(\widetilde{\mathbb{Y}}_t)^2, \tag{6}$$

$$\mathcal{L}_{D_s} = [1 - D_s(Y)]^2 + D_s(G(X))^2, \tag{7}$$

where $X$ denotes the low-resolution input, $Y$ denotes the high-resolution target, $\mathbb{Y}_t, \widetilde{\mathbb{Y}}_t$ denotes three consecutive frames of point clouds: $\mathbb{Y}_t = \{Y_{t-1}, Y_t, Y_{t+1}\}, \widetilde{\mathbb{Y}}_t = \{G(X_{t-1}), G(X_t), G(X_{t+1})\}$.

The corresponding adversarial losses for generator are:

$$\mathcal{L}_{\text{G},t} = [1 - D_t(\widetilde{\mathbb{Y}}_t)]^2, \tag{8}$$

$$\mathcal{L}_{\text{G},s} = [1 - D_s(G(X))]^2. \tag{9}$$

To encourage the output of generator lies close to the ground truth point clouds, we use Chamfer Distance (CD) to evaluate and penalize the distance between generator output $\tilde{Y}$ and ground truth $Y$

$$\mathcal{L}_{CD} = \frac{1}{|\tilde{Y}|} \sum_{\tilde{y} \in \tilde{Y}} \min_{y \in Y} ||\tilde{y} - y||_2^2 + \frac{1}{|Y|} \sum_{y \in Y} \min_{\tilde{y} \in \tilde{Y}} ||\tilde{y} - y||_2^2, \tag{10}$$

---

[1]This threshold can be set loosely (e.g. 0.1) without influencing final results too much, since in practice most of the mask values are very close to 1.0 or 0.0 after training.

[2]If an input point can only find 0~2 neighbors in the ground truth point cloud, then we will mark this point as "no need to upsample", as upsampling it is likely to introduce redundant points and violates point distribution.

where $y$ denotes point in $Y$, $\tilde{y}$ denotes point in $\tilde{Y}$, and $|\cdot|$ denotes the number of points in the corresponding point cloud. Compared to other advanced distance metrics, Chamfer Distance is simpler and thus much faster to evaluate.

Combining with the masking loss discussed in the previous section, the total loss for generator is defined as:

$$\mathcal{L}_G = \mathcal{L}_{\mathrm{G},t} + \mathcal{L}_{\mathrm{G},s} + \lambda_1 \mathcal{L}_{CD} + \lambda_2 \mathcal{L}_{\mathrm{mask}}. \qquad (11)$$

In practice we choose $\lambda_1$, $\lambda_2$ such that all loss terms have similar magnitude (roughly 0∼1).

**Stabilizing GAN's Training** We implement two techniques to stabilize the training process. The first one is spectral normalization proposed by Miyato et al. (2018), which constrains the Lipschitz constant of the discriminator by setting the spectral norm of weights to 1. We found this technique is crucial especially when training on the fluid dataset. We hypothesize the main reason is that the irregular distribution and complex dynamics of fluid particles can make discriminator prone to produce large gradients and thus make the training unstable. The second technique we adopt is label smoothing from Salimans et al. (2016), which softens the target label in the adversarial loss.

## 4 EXPERIMENT

### 4.1 IMPLEMENTATION DETAILS

**Dataset** We train and evaluate our model on two different datasets. The first dataset is generated from particle-based fluid dynamics simulation, where particles distribute and move under the physics governing equation. We use DFSPH (Bender & Koschier, 2015) solver from SPlisHSPlasH[3] and geometric assets from Ummenhofer et al. (2020) to generate the training data. We set the upsampling ratio to be $r = 8$ on the fluid dataset. The second dataset is from scanned human action - the MSR-Action3D dataset (Li et al., 2010), which consists of 567 Kinect depth videos of 20 action categories. In this dataset, points from two different frames do not necessarily have correspondence and number of points varies in different frames. We adopt the same preprocessing and splitting strategy from previous works (Liu et al., 2020; Fan et al., 2021b). The upsampling ratio is set to be $r = 16$ on the action dataset.

**Evaluation metrics** In addition to the Chamfer distance discussed in the prvious section, we introduce three other spatial distance metrics to further evaluate the model performance. The first metric is Earth Mover's Distance (EMD), which has been reported by previous work are more reliable and discriminative than Chamfer Distance in dense point clouds' evaluation (Liu et al., 2019a). EMD evaluates the distance between two clouds by solving a linear assignment problem. Other than EMD, we use Maximum Mean Discrepancy (MMD) to evaluate the spatial distribution difference between prediction and ground truth. MMD discriminates the point clouds by measuring their high-level similarity instead of difference within pairs of matched points. In practice we found MMD is most discriminative on dense point clouds, thus we only report MMD on the fluid dataset. We also report the Hausdorff distance (HD) following previous works in static point cloud upsampling task. The evaluation of the temporal coherence are depending on the specific tasks, so we will discuss them in the corresponding sub-sections.

**Baseline** To demonstrate the effectiveness of our proposed methods, we compare our model against two recent state-of-the-art works, PU-GCN (Qian et al., 2021) and Tranquil Clouds (Prantl et al., 2020). Tranquil clouds proposes the first learning-based method for upsampling point clouds in temporally coherent manner. Note that Tranquil Clouds uses mingling loss to encourage the points to spread and expand, this results in its much higher Chamfer Distance during evaluation, since Chamfer Distance favors upsampled points lying closer to their original input points. The second baseline PU-GCN is the state-of-the-art model on static point cloud upsampling.

(More details on metrics, baseline, and dataset generation can be found in Appendix A.2, A.3, A.4)

---

[3] https://github.com/InteractiveComputerGraphics/SPlisHSPlasH

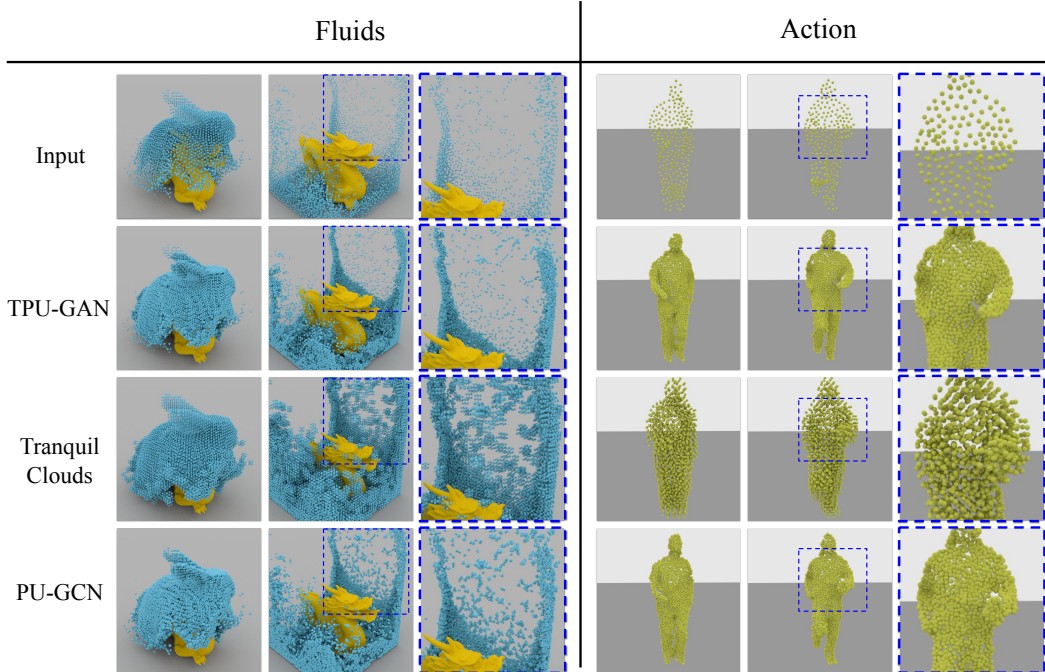

Figure 2: Qualitative comparison of TPU-GAN against other models on different datasets. To fully evaluate the temporal coherence of upsampled point cloud sequences, we refer readers to the rendered videos, which are available at: [Videos].

## 4.2 DENSE AND NON-UNIFORM POINT CLOUDS FROM PHYSICAL SIMULATION

**Advection consistency** In fluids, particles' motion are subject to the underlying governing equation, such that given the current position $\mathbf{x}_t$ and $\mathbf{v}_t$, there exists an continuous advection function: $\Psi : \mathbf{x}_{t+1} = \Psi(\mathbf{x}_t, \mathbf{v}_t)$ that advects particles from current state to next time step. Therefore we can evaluate the temporal coherence of generated sequences by checking if they are consistent with the underlying advection function. Given the ground truth data, we can use the kernel model in Smooth Particle Hydrodynamics (SPH) (Gingold & Monaghan, 1977) to approximate the value of advection function at arbitrary locations (for more details on SPH and kernel model, please refer to the Appendix A.5). Based on the approximated advection function $\Psi$ and the generator $G$, we define the advection consistency for generator as:

$$\mathcal{L}_{\text{adv}} = D\left(G(X_{t+1}), \Psi\left[G(X_t), V_t\right]\right), \tag{12}$$

where $X_t, V_t$ denotes the positions and velocities of particles at time step $t$, and $D(\mathcal{P}, \mathcal{Q})$ denotes the distance between two point clouds $\mathcal{P}, \mathcal{Q}$, which can be evaluated using the aforementioned spatial distance metrics.

| Model | Static | | | | Advection Consistency | | | |
|---|---|---|---|---|---|---|---|---|
| | CD↓ ×10⁻⁴ | EMD↓ ×10⁻² | MMD↓ ×10⁻⁵ | HD↓ ×10⁻² | CD↓ ×10⁻⁴ | EMD↓ ×10⁻² | MMD↓ ×10⁻⁵ | HD↓ ×10⁻² |
| TPU-GAN | 2.856 | **2.694** | 5.373 | 3.978 | 1.963 | 1.582 | 5.069 | 5.835 |
| TPU-GAN + vel | 2.964 | 2.722 | **5.058** | 4.009 | **1.632** | 1.434 | **3.545** | **5.593** |
| PU-GCN | **2.733** | 2.933 | 6.077 | **3.942** | 2.238 | 1.823 | 6.222 | 5.878 |
| Tranquil Clouds | 5.713 | 3.088 | 7.293 | 5.793 | 1.999 | **1.157** | 4.347 | 6.473 |

Table 1: Quantitative comparison of model performance on Fluids dataset. The *Static* column reports the per-frame difference between upsampled point clouds and ground truth. The *Advection Consistency* column reports the temporal coherence of two consecutive frames of upsampled point clouds.

**Comparison with other models** We tested out two versions of the proposed model - with or without using velocities as input features. The quantitative evaluation results are shown in the Table 1 and the visualization of sequence upsampled from real coarse fluid simulation are shown in the left column of Figure 2. Compared to PU-GCN which does not take temporal coherence into account, TPU-GAN improves the advection consistency without the information of displacements across frames, which demonstrates the effectiveness of temporal discriminator. TPU-GAN also outperforms PU-GCN on high-level distance metrics (EMD, MMD), which highlights the superiority of our proposed method in terms of reconstruction quality on irregular point clouds. Furthermore, we showcase that with velocity features, TPU-GAN is on par with Tranquil Clouds in terms of temporal coherence, while improves the static upsampling quality on all the metrics. Note that Tranquil Clouds is strongest when using EMD to evaluate advection consistency, in part because it is trained with a temporal loss that is essentially equivalent to evaluating the advection consistency using EMD, with advection function approximated by EMD assignment instead of SPH's kernel interpolation.

## 4.3 REAL-WORLD SCANNED POINT CLOUDS

| Model | Frame | | | Sequence |
|-------|-------|-------|-------|----------|
| | $CD\downarrow$ $\times10^{-3}$ | $EMD\downarrow$ $\times10^{-2}$ | $HD\downarrow$ $\times10^{-1}$ | $PL\downarrow$ |
| TPU-GAN | 1.231 | **5.720** | **1.500** | **0.145** |
| PU-GCN | **1.173** | 5.884 | 1.503 | 0.156 |
| Tranquil Clouds (8x) | 2.262 | 5.987 | 1.669 | 0.201 |
| Tranquil Clouds (16x)[4] | 2.483 | 6.243 | 1.697 | 0.224 |

Table 2: Quantitative comparison of different models on MSR-Action3D dataset.

| Model | # Frames | Accuracy (%) $\uparrow$ |
|-------|----------|-------------------------|
| PointNet++ | 1 | 61.75 |
| PSTNet | 3 | **77.78** |
| SA & MLP | 3 | 42.39 |
| TempoDis Feature + SA & MLP | 3 | 74.07 |

Table 3: MSR-Action 3D classification results using Temporal Discriminator's feature.

**Perceptual loss for point cloud sequences** For real-world scanned data, we often don't know the corresponding points in the next frame given the current frame. Therefore it is difficult to directly compare the motion of upsampled sequence and target sequence. Inspired from previous video generation works which use pretrained I3D network (Carreira & Zisserman, 2018; Unterthiner et al., 2018) to evaluate the spatio-temporal quality of generated videos at sequence level, we use a pretrained point cloud classifier to extract sequence-level spatio-temporal features. The pretrained classifier we used here is PSTNet (Fan et al., 2021b), the state-of-the-art model on point cloud sequence classification. We use the feature map from the middle of the network as the extracted features (more details can be found in Appendix A.3), and based the on the extracted features we define the perceptual loss (PL) for point cloud sequences as:

$$\mathcal{L}_{\text{PL}} = \left\lVert \phi(\mathbb{Y}) - \phi(\tilde{\mathbb{Y}}) \right\rVert_1, \tag{13}$$

where $\mathbb{Y} = \{Y_1, Y_2, \ldots, Y_T\}$ denotes the reference sequence, $\tilde{\mathbb{Y}} = \{\tilde{Y}_1, \tilde{Y}_2, \ldots, \tilde{Y}_T\}$ denotes the predicted sequence and $\phi$ denotes the feature extractor.

**Comparison with other models** The quantitative evaluation result on MSR-Action 3D dataset is shown in the Table 2 and visualization of upsampled point clouds is shown on the right column of the Figure 2. As indicated in the quantitative result, TPU-GAN can improve the upsampling quality in both frame-level and sequence-level under a scenario that no point correspondence annotation is available.

**Self-supervised temporal feature learning** To further evaluate the representation learned by temporal discriminator, we apply it as a feature extractor for classification task. We freeze all the layers before the last set abstraction layer (SA) and MLP, then test the network's performance on action classification. In Table 3, we report the video-level accuracy for three supervised methods and classifier based on feature extracted by temporal discriminator. By leveraging spatiotemporal feature extracted by the temporal discriminator, the classifier outperforms PointNet++ that only utilizes single-frame features and outperforms model which uses only the set abstraction layer and MLP. PSTNet still leads the performance, as it is trained in a fully-supervised way (i.e. all parameters are trained on the classification task) and contains more parameters (PSTNet: 7.4M, TempoDis: 0.88M).

---

[4]We tested two version of Tranquil Clouds - 8x upsampling version (with 256 input points) following the original paper, and a 16x upsampling version (with 128 input points).

Table 4: Ablation on model architecture

| Model | Frame | | | Sequence |
|---|---|---|---|---|
| | CD↓ $\times 10^{-3}$ | EMD↓ $\times 10^{-2}$ | HD ↓ $\times 10^{-1}$ | PL↓ |
| Full model | 1.231 | **5.720** | 1.500 | **0.145** |
| *w/o* SpatialDis | 1.356 | 6.021 | 1.508 | 0.156 |
| *w/o* TempoDis | **1.185** | 5.821 | **1.458** | 0.155 |
| Input two frames | 1.206 | 5.743 | 1.489 | 0.149 |
| Input five frames | 1.295 | 5.946 | 1.535 | 0.159 |

Figure 3: Comparison of distribution of points' top $0.1\%$ furthest distance

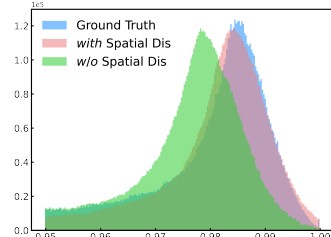

## 4.4 ABLATION STUDY

**Architectural choice**   We investigate the influence of our architectural choice to the model performance on the MSR-Action 3D dataset. The result is shown in the Table 4. The result indicates that: first, increasing the frame number of point clouds exposed to the temporal discriminator can improve the the sequence level accuracy. However too long input sequence ("Input five frames") will deteriorate the frame-level upsampled quality. As it will make points cluster around input points and exhibit halo artifacts. (See Figure 10 in the Appendix for visualization) Moreover, we find that spatial discriminator is important for the non-uniform dense point cloud upsampling. The upsampled points will shrink towards the center of the point cloud without spatial discriminator. We calculates the top $0.1\%$ furthest distances between every pair of points in the upsampled patches from fluid dataset, and we normalize all the distances using the furthest distance in ground truth. As shown in the Figure 3, without spatial discriminator, the upsampled point clouds' furthest distance distribution will shift to the left.

**Adaptive upsampling**   We analyze the point distribution in the upsampled results from different methods by comparing their density to the ground truth. The analysis was conducted on the fluid data, where particle density is a crucial physical quantity that dominates the dynamics of incompressible fluids. We evaluate the particle number density following the definition in the SPH model, and the result is shown in the Figure 4. For ground truth data, the density distribution has two sharp peaks, which corresponds to particles near surface area and particles in the central part respectively. For upsampled point clouds, all the distribution has a smoothed bell shape, while our method produce a sharper density distribution with a mode $\eta = 7.4$ more close to the ground truth ($\eta_1 = 6.3, \eta_2 = 8.4$). This demonstrates the adaptive masking module in the generator effectively learns to suppress unnecessary upsampling and approximates the target density distribution, while other methods have the density distribution shifted to the right.

**Model size**   In Table 5 of Appendix, we report the number of trainable parameters for concurrent learning-based upsampling models. Despite the dual discriminator setting, our model is still much smaller than PU-Net and Tranquil Clouds which in turn makes it more efficient during inference.

## 5 CONCLUSION

In this work, we propose TPU-GAN, a GAN framework for upsampling dynamic point cloud sequences. Our proposed method is able to learn useful spatiotemporal representation for both generative task and discriminative task from raw point cloud sequences. Through experiments on different data sources, we demonstrate the effectiveness of our method on the generative modelling of point cloud sequences.

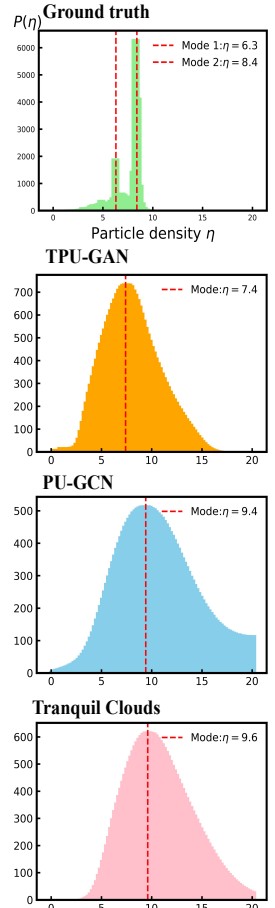

Figure 4: Quantitative comparison of density distribution of different upsampling methods on Fluid dataset.

ACKNOWLEDGEMENT

This work is supported by the start-up fund provided by CMU Mechanical Engineering.

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

# A APPENDIX

## A.1 MODEL DETAILS

We implement our model in PyTorch 1.7.1. All the training and experiments are run on a platform equiped with a single GTX-1080Ti. We train all our model for 100k gradient updates using Adam optimizer (Kingma & Ba, 2017), which took approximately 30 hours on the Fluid dataset and 15 hours on the MSR-Action 3D dataset. The detail of our model architecture are listed below:

- For all the GCN, set abstraction and flow embedding layer, we use max pooling as aggregation function.

- The feature extractor in the **Generator** comprises one layer of GCN and two layers of Inception DenseGCN. At the GCN layer, we construct neighborhood by searching each points' *20* nearest neighbors based on the point position and use a MLP with hidden units: $[3, 32, 128]$ to encode the point features. For Inception DenseGCN we follow the design and hyperparamter choice in the original PU-GCN paper (Qian et al., 2021).

- The masking and upsampling module in the **Generator** has two layer of GCN and a $[128, 64, c]$ MLP, where $c = 3r$ for upsampling module, $c = 1$ for masking module and $r$ is the upsampling ratio. The first layer of GCN operates on the *12* nearest neighbors and the second layer operates on the *4* nearest neighbors. Each GCN layer contains a $[128, 32]$ bottleneck linear layer and a $[32, 64, 128]$ MLP to update point features.

- We follow the design of PointNet++ to build the **Spatial Dsicriminator**. Concretely, we use three set abstraction layers:[5] $(0.25N, 32, [3, 64, 64, 128])$, $(0.125N, 32, [128, 128, 128])$, $(0.0625N, 32, [128, 128, 256])$, (all, $[256, 256, 512]$) and a $[512, 256, 64, 1]$ MLP (with $N$ the number of input points).

- The **Temporal Discriminator** also adopts a hierarchical structure that contains two set abstraction layers: $(0.25N, 64, [3, 64, 64, 128])$, $(0.125N, 32, [128, 128, 256])$, followed by two flow embedding layers:$(0.125N, 32, [256, 256, 256])$, $(0.125N, 32, [256, 256, 256])$, and finally a set abstraction layer: (all, $[256, 256, 512]$) and a MLP: $[512, 256, 64, 1]$.

Figure 5: Schematic of flow module. The flow module takes $T$ frames of point cloud features $\{F_1^{(0)}, F_2^{(0)}, \ldots, F_T^{(0)}\}$ as input. Inside each layer, two consecutive frames of point clouds' features $(F_t^{(l)}, F_{t+1}^{(l)})$ are mixed via a shared flow embedding layer, then the output features $F_t^{(l+1)}$ are stored on the former point cloud $X_t$. For $T$ input frames, the flow module will have a depth of $T - 1$ ($T - 1$ different flow embedding layers).

---

[5]$(M, K, [h_1, \ldots, h_l])$ means that we use Furthest Point Sampling (FPS) to sample $M$ local regions, each with $K$ sampled points, and then use a $[h_1, \ldots, h_l]$ MLP to update point features.

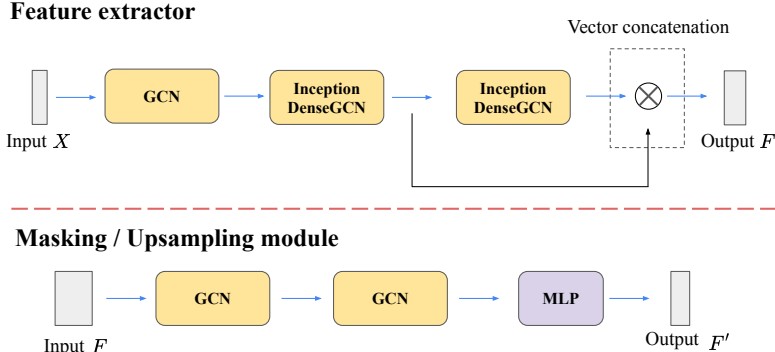

Figure 6: **Top**: Schematic of feature extractor in the generator. The feature extractor comprises three layers of Inception DenseGCN and concatenates features from different scales to derive the final output. **Bottom**: Schematic of masking/upsampling module in the generator. Following the design of NodeShuffle module in PU-GCN, inside masking/upsampling module, latent features are first aggregated from local neighborhood via GCN and then decoded into output by a MLP.

Comparison on the number of parameters of different models is shown below.

| Model | PU-Net | PU-GAN | PU-GCN | Tranquil Clouds | TPU-GAN |
|---|---|---|---|---|---|
| # Param. (M) | 3.26 | 2.74 | 0.30 | 4.87 | 1.74 |

Table 5: Comparison on model size of TPU-GAN vs other state-of-the-art models

## A.2 DATASET

**Fluid** We simulate 24 fluid sequences with random initialization of fluid block position and scene configuration, each sequence contains 200 snapshots with a time interval of 0.005s and particle radius of 0.0125. Each snapshot contains roughly $100k \sim 200k$ particles. We use 20 sequences as training data, and test on the rest 4 sequences. We adopt the patch training strategy, where we sample patches containing 8192 particles from the fluid field and uniformly downsample them with farthest point sampling to create input-target pairs for training.

**MSR-Action 3D** We follow the description from Liu et al. (2020) [6] to process the raw data into point clouds. We uniformly downsample the processed data with farthest point sampling to create low-resolution input and input the whole frame to the network. When evaluating metrics on MSR-Action 3D, we normalize the point clouds to lie inside the unit ball $[-1, 1]$ to balance the influence of point clouds of different classes. On MSR-Action 3D, we observe that masking module has less influence on the final output and only very few points are masked.

## A.3 METRIC

**Earth mover distance** We use Liu et al. (2019a)'s EMD implementation[7] to evaluate the loss. This implementation is based on auction algorithm and thus is consistent with the EMD code used in the original Tranquil Clouds paper. In fact, our evaluation of Tranquil Clouds on the fluid dataset also roughly matches the results reported in their original paper ($3.09 \times 10^{-2}$ in our paper, $3.46 \times 10^{-2}$ in original paper, the difference is likely caused by our fine-tuning on the pretrained weights and different implementation of EMD). To match the EMD metric reported in Prantl et al. (2020), a convergence threshold of 0.03 is adopted on the Fluid dataset. For MSR-Action 3D, we use a threshold of 0.002.

**Maximum Mean Discrepancy** We use the GeomLoss library[8] to evaluate the MMD, with a Gaussian kernel and bandwidth $\sigma = 0.01$. The point clouds' coordinates are normalized to $[0, 1]$ before evaluation.

---

[6]https://github.com/xingyul/meteornet/tree/master/action_cls
[7]https://github.com/Colin97/MSN-Point-Cloud-Completion
[8]https://github.com/jeanfeydy/geomloss

**Perceptual loss for point cloud sequence**    We evaluate the perceptual loss for action dataset based on features from the third convolution layer of PSTNet. As shown in the original paper (Fan et al., 2021b), the convolution layer output higher activation on the salient motion. This motivates us to compare the activation of reconstructed point clouds against the ground truth, which provides a measurement for the similarity of two sequences in terms of motion. In practice we found that using activation from different middle layers (layer $2, 3, 4$) produce similar results. To make the extracted feature permutation invariant to the input points, we conduct a global top-k pooling (with $k = 128$) before calculating the loss.

## A.4    Baseline

**Tranquil Clouds**    We use the publicly available source code[9] released by the authors to carry out experiment. On the fluid dataset, we find it unnecessary to retrain the model since the fluid data in both papers are essentially generated from similar volume-conserving SPH simulator, so we just fine-tune the pretrained weights provided by the authors on our own fluid dataset for 5 epochs. For action dataset, we explore two strategies to train the model. The first is to train the model from scratch without temporal loss, since there is no point correspondence information in the dataset. However this makes Tranquil Clouds perform much worse in terms of temporal coherence. The second is directly using the pretrained weights which preserves temporally coherent latent space. In general we found the latter provides much better temporal coherence thus we stick with pretrained weights to conduct experiment.

**PU-GCN**    We implement PU-GCN in PyTorch and compare our implemented version with the official TensorFlow version released by the authors[10]. We found they have similar performance. As our training and evaluation pipeline was mostly implemented in PyTorch, so for convenience we use PyTorch version of PU-GCN throughout the experiment.

## A.5    Kernel model in SPH

A common method to solve the Navier-Stokes equation and discretize fluids under the Lagrangian framework is Smooth Particle Hydrodynamics (SPH) (Gingold & Monaghan, 1977). In SPH, physical quantities at an arbitrary point in the space are approximated by the quantities of nearby particles.

Conretely, an arbitrary scalar (or vector) field $A(\mathbf{r})$ at location $\mathbf{r}$ can be represented by a convolution:

$$A(\mathbf{r}) = \int A(\mathbf{r}') W(|\mathbf{r} - \mathbf{r}'|, h) \, dV(\mathbf{r}') \tag{14}$$

$$\approx \sum_{j \in \mathcal{N}(i)} A(\mathbf{r}_j) \frac{m_j}{\rho_j} W(|\mathbf{r}_i - \mathbf{r}_j|, h), \tag{15}$$

$$\text{and} \quad \rho_i = \sum_{i \in \mathcal{N}(i)} m_j W(|\mathbf{r}_i - \mathbf{r}_j|, h), \tag{16}$$

where $W$ is weighting function or smooth kernel as defined in SPH, $h$ is the cutoff, which defines the range of particles to be considered, $V(\mathbf{r}')$ denotes the volume at $\mathbf{r}$, $\mathcal{N}(i)$ denotes the neighbor particles of particle $i$ that are within the cutoff, $\rho$ denotes the particle number density and $m$ denotes the mass of each fluid particle.

Using this model, we can approximate the advection $\Psi(\mathbf{r})$ at arbitrary location $\mathbf{r}$ by:

$$\Psi(\mathbf{r}) = \sum_{j \in \mathcal{N}(i)} \Psi(\mathbf{r}_j) \frac{m_j}{\rho_j} W(|\mathbf{r}_i - \mathbf{r}_j|, h), \tag{17}$$

with,

$$\Psi(\mathbf{r}_j) = \mathbf{r}_j^{(t+1)} - \mathbf{r}_j^{(t)}, \tag{18}$$

where $\mathbf{r}_j^{(t)}$ denotes the position of particle $j$ at time step $t$. Throughout the experiment we use cubic spline kernel function as the smooth kernel $W(\cdot)$.

---

[9]https://gitlab.com/Prantl/NeuralParticles
[10]https://github.com/guochengqian/PU-GCN

## A.6 VISUALIZATION AND STUDY ON THE MASKING MODULE

In this section we present visualization on the learnable masking module from different perspectives. Figure 7 shows the effect of masking module on the very same network. Figure 8 visualizes the prediction of masking module on randomly selected fluid patches. Figure 9 shows the superiority of using proposed masking loss (equation 4) to train the masking module. Without the supervision of masking loss, the mask module will suppress the upsampling too much, which results in clustered points and halo structures.

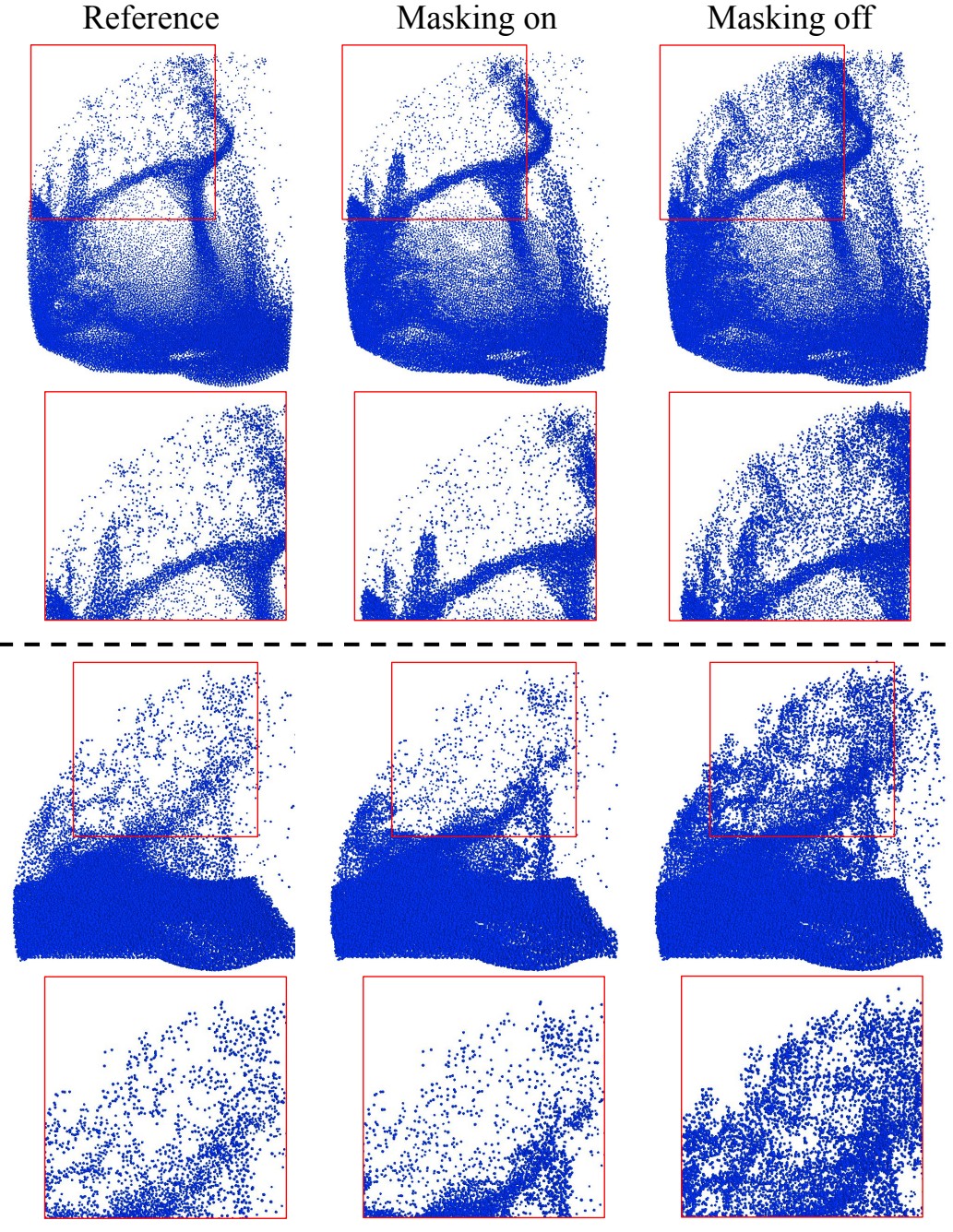

Figure 7: Visualization of the effect of masking module on fluid dataset (zoom in for better view). To "turn off" the masking module, we set all the values in the mask array predicted by the masking module to one.

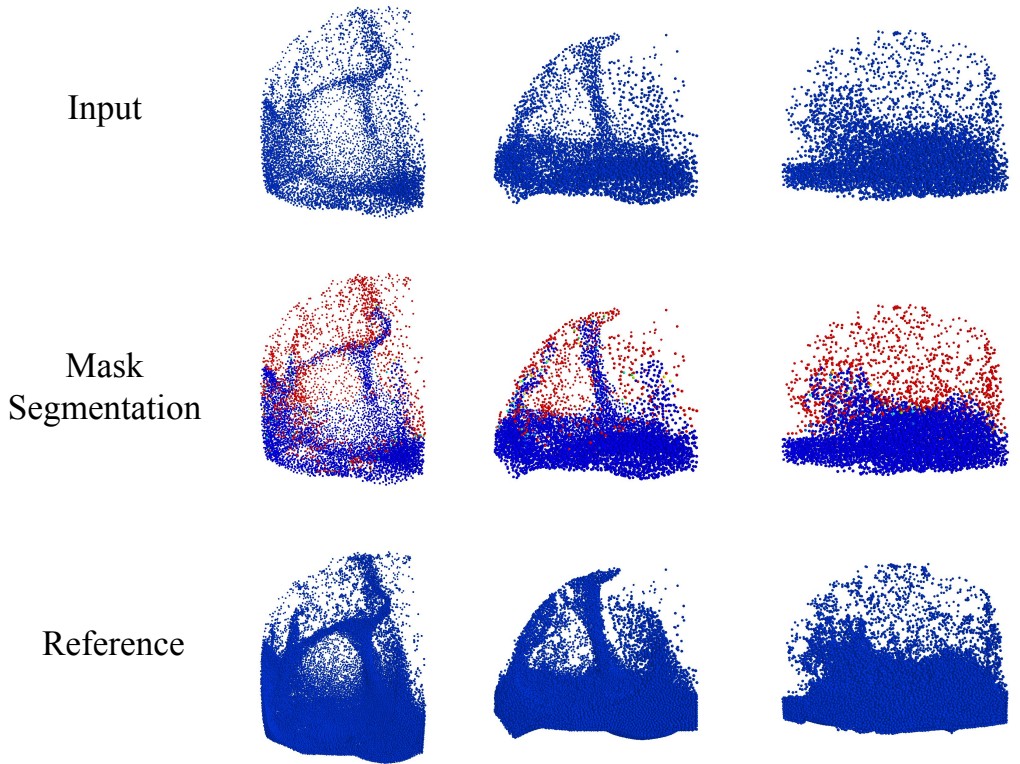

Figure 8: Visualization of the masking module's prediction (zoom in for better view). Given a sparse input, masking module will predict a value for each point to indicate if it needs to be upsampled. Here we normalize all the mask value to $[0, 1]$, and then color each point with their corresponding mask, red indicates mask value for this point is close to zero (no need to upsample), blue indicates mask value is close to one (need to be upsampled).

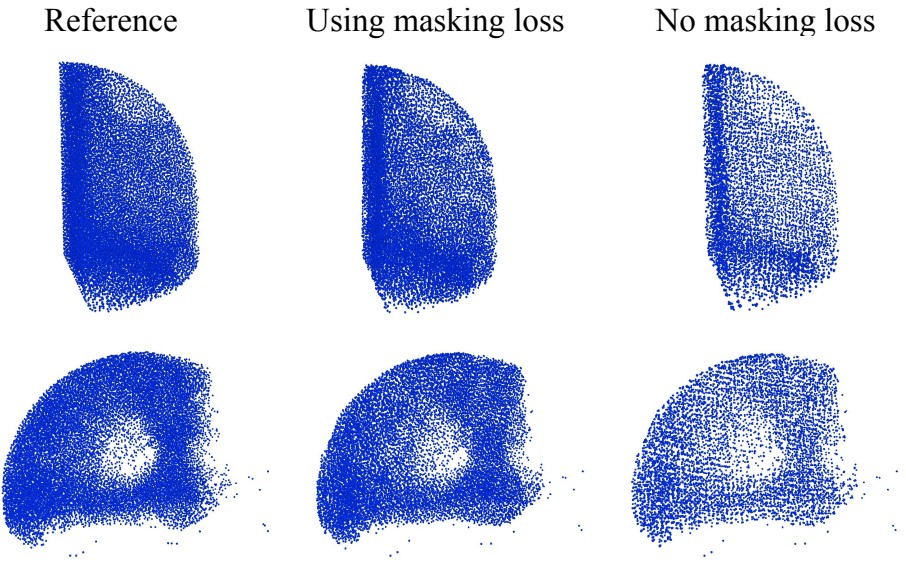

Figure 9: Visualization of the upsampling results based on different masking module training methods (zoom in for better view). When not using masking loss, we remove the non-differentiable quantization process (equation 3) and directly train the masking module based on gradient from other losses (adversarial loss, Chamfer distance). Without masking loss' supervision, the upsampled results contain clustered points and halo structures.

## A.7 VISUALIZATION OF ABLATION

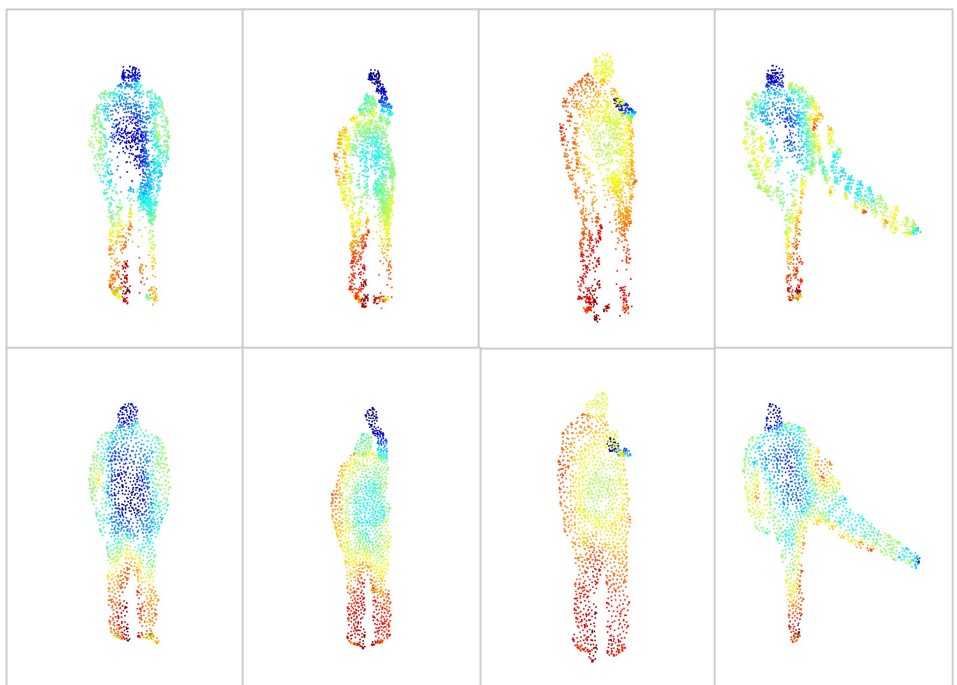

Figure 10: Visualization of snapshots from different sequences in MSR-Action 3D dataset, color indicates the depth of point. **Top row**: results from TPU-GAN with temporal discriminator taking **5** input frames; **Bottom row**: results from TPU-GAN with temporal discriminator taking **3** input frames. The results at the top row exhibit halo structures.

## A.8 VISUALIZATION OF POINT CLOUDS GENERATED DURING TRAINING PROCESS

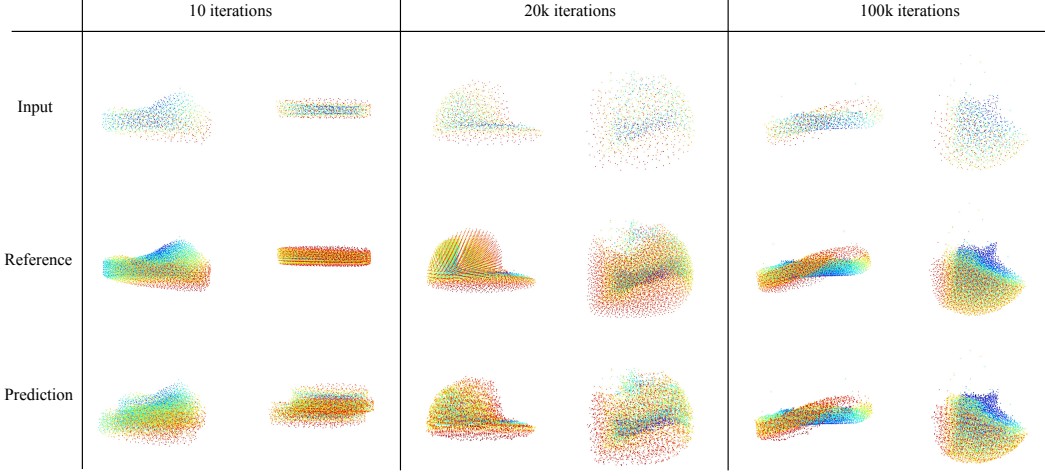

Figure 11: Visualization of fluid point cloud generated from model during the training process. The input and reference point clouds are randomly selected and cropped from the testing dataset. Point color indicates depth.

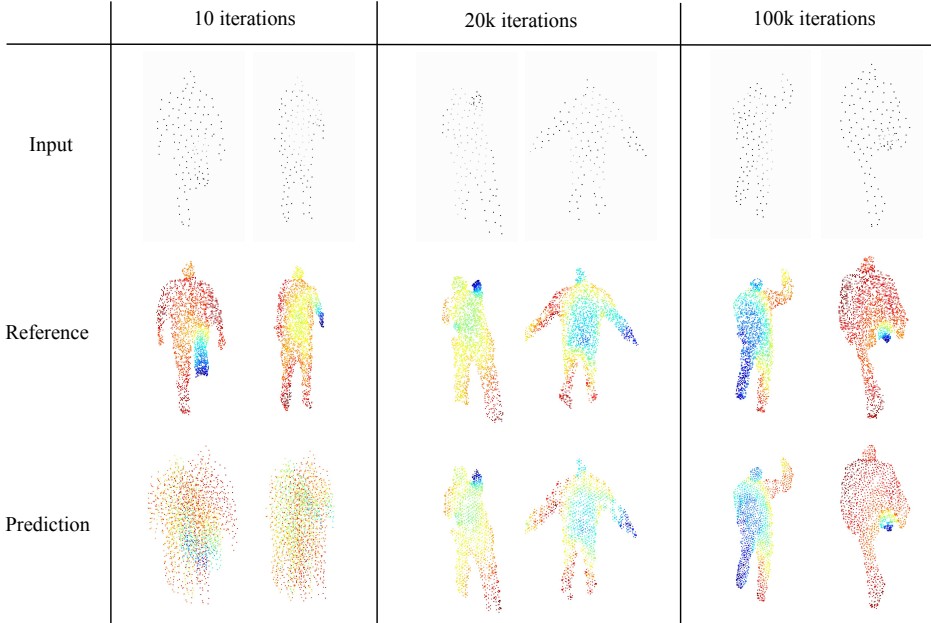

Figure 12: Visualization of MSR-Action3D point cloud generated from model during the training process. The input and reference point clouds are randomly selected from the testing dataset. For better visualization, the input points are shown in grey scale. Point color indicates depth.

## A.9 SCENE-FLOW ESTIMATION

We apply our proposed method to a self-driving dataset (Kitti-scene flow) to demonstrate the effectiveness of our model in real-world self-driving application. We use the Kitti point cloud scene flow dataset proposed by Liu et al. (2019b). To mimic real world scenario where data are very sparse, we downsample the original point clouds (around 30000 - 40000 points per frame) to obtain sparse point clouds (2048 points per frame). We apply different upsampling networks to upsample the sparse point clouds (visualization is shown in Figure 13). Then we use upsampled point clouds as input data for a scene flow prediction network (pre-trained FlowNet3D (Liu et al., 2019b)).

To quantitatively measure model's performance under different conditions, we adopt two metrics to evaluate the accuracy of predicted scene flow - end point error (EPE 3D), which evaluates the L2 distance between ground truth and prediction, and 3D outliers, which evaluates how many predictions are extremely deviated from ground truth (>0.3m or >5%). Since data is very limited here (only 150 pairs of two consecutive frames are available), we follow previous works' evaluation protocol (Li et al., 2019; Qian et al., 2021; Liu et al., 2019b), all the upsampling networks are pretrained (on the fluid dataset) instead of training from scratch on Kitti scene data.

As shown in the Table 6, the performance of scene flow prediction network drops significantly when input points are sparse, while upsampling networks can help mitigate the performance degradation. Particularly, TPU-GAN outperforms other networks on both metrics. This demonstrates the superiority of our proposed method and its potential to self-driving application.

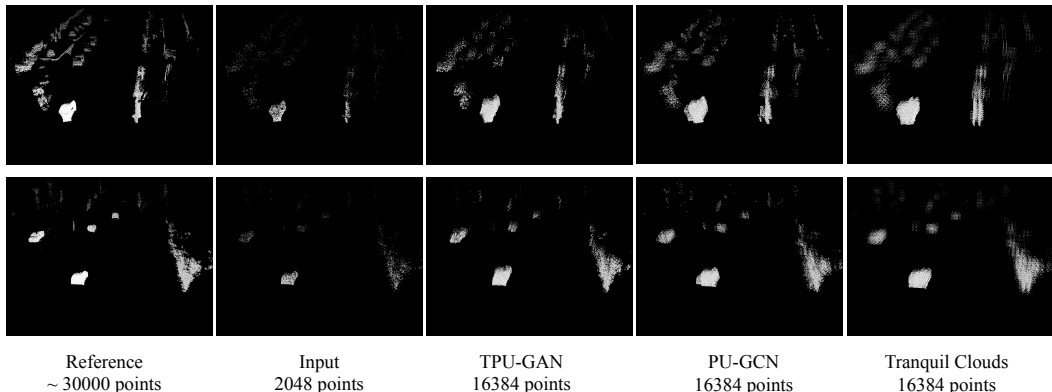

|  Reference | Input | TPU-GAN | PU-GCN | Tranquil Clouds |
| ~ 30000 points | 2048 points | 16384 points | 16384 points | 16384 points |

Figure 13: Visualization on Kitti-scene flow dataset. (Zoom in for better view)

| Input source | EPE 3D↓ | 3D outliers↓ |
|---|---|---|
| Full (16384 points) | 0.1974 | 17.1% |
| Part (2048 points) | 0.2577 | 27.5% |
| TPU-GAN | **0.2243** | **19.8%** |
| PU-GCN | 0.2362 | 23.0 % |
| Tranquil Clouds | 0.2392 | 22.5% |

Table 6: Scene flow estimation performance of pre-trained FlowNet3D on different input sources. *Full* means using all available points as input, following the setting in Liu et al. (2019b). *Part* means using the downsampled sparse data.

