# OpenReview forum: "TPU-GAN: Learning temporal coherence from dynamic point cloud sequences"
_ICLR.cc/2022/Conference — ICLR 2022 Poster_

### Official Review · Reviewer_evrR · 2021-10-29

**Correctness:** 3
**Technical Novelty And Significance:** 3
**Empirical Novelty And Significance:** 3
**Recommendation:** 6
**Confidence:** 4

**Main Review:**

strengths
- TPU-GAN can learn temporal information from sequential point clouds and the experiments show that TPU-GAN can learn the underlying temporal coherence.
- TPU-GAN can determine for each point whether it should be upsampled or not, and perform upsampling only for the points that need upsampling.

weaknesses
- There is no ablation study to confirm whether the function to determine for each point whether it should be upsampled or not is realized by the masking module.
- The reason for employing the spatial discriminator should be explained.  I could not understand the reason why calculating temporal discriminator loss and spatial discriminator loss because temporal discriminator seems to include spatial discriminator in its structure.

Comment
- The explanations of (b) and (c) in Figure 1 is reversed.
- I think "Table 4.3" in the fourth line of the subsection "Self-supervised temporal feature learning" in section 4.2 is a mistake for "Table 3".
- In the fourth line of "Self-supervised temporal feature learning" in section 4.2 "Table 4.3" may be "Table 3".


**Summary Of The Paper:**

This paper introduced a method of point cloud upsampling using GAN, which can learn the underlying temporal coherence from a point cloud sequence.  TPU-GAN can determine for each input point whether it should be upsampled or not by introducing a masking module into the conventional point clouds upsampling model. In addition, TPU-GAN can learn the underlying temporal coherence using temporal discriminator that takes temporal sequential point clouds as input. The experiments are evaluated with particle-based fluid dynamics simulation dataset and scanned human action dataset. In the comparison,  the method proposed in this paper outperformed PU-GCN and Tranquil Clouds. Moreover, to confirm the performance variation with the number of input frames and to confirm the necessity of the both temporal discriminator and spatial discriminator in the learning, some other experiments are conducted.


**Summary Of The Review:**

Please provide a short summary justifying your recommendation of the paper.  TPU-GAN is well designed to learn the underlying temporal coherence from a point cloud sequence, but additional experiments and additional explanations about the model and modules are necessary.

---

> ### Author Response · Authors · 2021-11-16
> **Response to reviewer evrR**
>
> We thank the reviewer for the constructive comment. We would like to address concerns as below.
>
> ### Weaknesses:
>
> * Q: **There is no ablation study to confirm whether the function to determine for each point whether it should be upsampled or not is realized by the masking module.**
>
> Thanks for the suggestion. We have added visualizations of the effect of the masking module in section A.6 “Visualization and study on masking module” of the revised manuscript. The influence of masking module on the upsampling network is shown in Fig.7. We refer reviewer to our revised manuscript for more details.
>
> * Q: **The reason for employing the spatial discriminator should be explained.**
>
> It is true that spatial discriminator has a very similar architecture as temporal discriminator, but in practice we found spatial discriminator and temporal discriminator have quite different effects on model’s performance (especially when the length of input sequence is greater than 3). As shown in Table 4, dropping the spatial discriminator will deteriorate model’s performance on frame-level metrics.
>
> In addition, on dense point cloud datasets like fluids, we observed that upsampled point clouds will shrink towards the center (as shown in Fig.3) without the spatial discriminator. We hypothesize this is because Chamfer distance encourages points to locate closer to the dense regions, where it could find more correspondences with respect to points in the ground truth, while spatial discriminator can help alleviate this.
>
> ### Comment:
>
> * Q:
>
> **The explanations of (b) and (c) in Figure 1 is reversed.**
>
> **"Table 4.3" in the fourth line of the subsection "Self-supervised temporal feature learning" in section 4.2 is a mistake for "Table 3"**
>
> **In the fourth line of "Self-supervised temporal feature learning" in section 4.2 "Table 4.3" may be "Table 3".**
>
> We appreciate the reviewer for pointing out these typos. They are now fixed in the revised manuscript.

---

> > ### Comment · Reviewer_evrR · 2021-11-30
> > **Post-Rebuttal Comment**
> >
> > My concerns about the discriminators and masking module were solved by your answer and additional experiments. However, I remain my score because of the limitation of the contribution of your paper.

---

### Official Review · Reviewer_gFvn · 2021-11-01

**Correctness:** 4
**Technical Novelty And Significance:** 2
**Empirical Novelty And Significance:** 3
**Recommendation:** 6
**Confidence:** 4

**Main Review:**

Strengths:

1. Avoiding explicit supervision on the point displacements (scene flows)  is an important and big challenge for point cloud sequence modelling.  The direction that spatial-temporally upsamples point cloud sequences without scene flow supervision is correct and promising.

2. The proposed method that employs a GAN framework for the super-resolution of dynamic point cloud sequence is technically sound.

3. The proposed method is evaluated on two very different datasets and experiments demonstrates the effectiveness of the proposed methods.

Weaknesses:

1. Although addressing an interesting problem, the novelty and contribution seem a bit limited. GAN frameworks, such as PU-GAN,  have been used for point cloud upsampling. Modules, such as Set Abstraction (PointNet++) and Flow Module (FlowNet3D), are also widely used for static point cloud and point cloud sequence processing.  Combining these techniques to solve a new problem seems not that novel.

2. The learnable masking module, which adaptively upsamples points, is a difference from previous works. It cloud be better to provide the visualization for this module to show what it learns.

3. Self-driving is a more important application for point clouds. It cloud be better to apply the proposed method on self-driving car datasets to demonstrate its effectiveness.

4. Upsampling three frames is somewhat short. It could be interesting to investigate the influence of sequence length on this task.

5. A few related works are missing. For example, [a] interpolates frames into point cloud sequences, which can be seen as temporally upsampling.  [b] and [c] also learn Spatio-temporal representations of point cloud sequences without point displacement supervision.

[a] PointINet: Point Cloud Frame Interpolation Network (AAAI21)

[b] Point 4D Transformer Networks for Spatio-Temporal Modeling in Point Cloud Videos (CVPR21)

[c] Anchor-Based Spatio-Temporal Attention 3-D Convolutional Networks for Dynamic 3-D Point Cloud Sequences (TIP21)

**Summary Of The Paper:**

This paper proposes a GAN framework for super-resolution on the dynamic point cloud sequences.
The main purpose is to avoid explicit supervision on the point displacements (scene flows) for this task.
Specifically, the Generator aims to produce super-resolution points for each individual frame. Then, a Temporal Discriminator and a Spatial Discriminator are used to measure the confidence of temporally coherent and spatially coherent, respectively.
Experiments on fluid particles and 3D scanned human action that the proposed upsampling model can produce spatially and temporally point cloud sequences consistent with target data without explicit supervision on the point displacements.

**Summary Of The Review:**

The paper presents an interesting problem. The proposed method is effective. However, the novelty seems a bit limited. Some experiments and related works are missing.

---

> ### Author Response · Authors · 2021-11-16
> **Response to reviewer gFvn**
>
> We thank the reviewer for the insightful comment. Below we would like to address your concerns.
>
> ### Weaknesses:
>
> * Q: **The novelty and contribution of this work.**
>
> Despite GAN has already been studied in the generative modeling of point clouds.  To the best of our knowledge, we propose the first generalized spatial-temporal GAN framework for 3D point cloud sequences, and we demonstrate that our proposed framework can be applied to different data sources without heavy tweaking. In addition, we propose an efficient adaptive upsampling strategy for non-uniform point clouds.
>
> * Q: **The learnable masking module, which adaptively up-samples points, is a difference from previous works. It cloud be better to provide the visualization for this module to show what it learns.**
>
> We appreciate the reviewer for this constructive suggestion. We have now added visualization in the revised manuscript. Please refer to section A.6 “Visualization and study on the masking module” in the revised manuscript for more details. We analyze the influence of masking module on the very same network (**Fig.7**), the visualization of predicted mask (**Fig.8**) and comparison of using different ways to train masking module (**Fig.9**).
>
> * Q: **It cloud be better to apply the proposed method on self-driving car datasets to demonstrate its effectiveness.**
>
> Thanks for the suggestion, self-driving task is indeed of great practical importance. We have now added an experiment on self-driving dataset in the section A.9 of revised manuscript. Concretely, we applied our pretrained model to upsample real-world scanned data and show that our method can improve the accuracy of downstream task like scene flow regression when point cloud data is very sparse and limited. Please refer to the revised manuscript for more details.
>
> * Q: **Upsampling three frames is somewhat short. It could be interesting to investigate the influence of sequence length on this task.**
>
> The effect of the input sequence length of the temporal discriminator is presented in Table 4 and Figure 10. In general, we observed that too long input sequence will make network produce points that are clustered around motion trajectories and exhibit halo artifacts.
>
> * Q: **A few related works are missing.**
>
> Thanks for the references. We have now added discussion about them in the revised manuscript.

---

> > ### Comment · Reviewer_gFvn · 2021-11-23
> > **Post-Rebuttal**
> >
> > Thank the authors for their responses, which address the majority of my questions. However, it seems that the method is not good at modelling long sequences, which somewhat limits the contribution.  Anyway, the paper presents an interesting problem. I therefore remain my score.

---

### Official Review · Reviewer_7Wbg · 2021-11-01

**Correctness:** 4
**Technical Novelty And Significance:** 2
**Empirical Novelty And Significance:** 3
**Recommendation:** 6
**Confidence:** 3

**Details Of Ethics Concerns:**

-

**Main Review:**

The paper proposes TPU-GAN,  an approach for point cloud generation concerning temporal coherences. In addition, the authors introduce a new trainable mask for controlling the distribution of sampled points. Both methods are well motivated. The authors provide experiments investigating various aspects of the proposed approaches. While the results seem to be reasonable, there are several concerns stated below.

Major comments:

1. I find methods to be well explained and experimental part to be sufficient. The authors' results are reasonably better than others analyzed methods. Nevertheless, I find that the main contribution, TPU-GAN, is based on several well-known ideas or techniques with small, in my opinion, novelty. Here is how I understand the novelty of the work: there already exist approaches that use GAN, for example, PU-GAN and PU-GCN. The novelty here is to use a GAN that incorporates learning temporal coherence. However, in my opinion, the background behind the TPU-GAN is based on existing approaches: feature extractor follows PU-GCN,  Temporal Embedding Layers used in the architecture are from work [4] with a small difference in input features ($ F_t, F_{t + 1}$). Spatial and temporal loss functions are common for GANs. In addition, the idea of using the triple moment($t - 1, t, t + 1$) for training the model appears in the paper [5].  Please, give clear arguments to show the significant novelty of TPU-GAN.

2. It’s claimed in the text that “Tranquil Clouds requires point correspondence across frames to evaluate the first-order and second-order finite difference loss, which our method (TPU-GAN) does not require during the training phase.” However, TPU-GAN’s loss function still relies on ground truth in $L_{CD}$ . Please, clarify what exactly was meant by this statement?

Minor comments:

1. I recommend the authors look through the recent work "Sequential point cloud upsampling by exploiting multi-scale temporal dependency" by Wang K [3] and discuss the similarities and differences with the proposed TPU-GAN. In this paper, the authors also proposed an approach for non-uniform sparse upsampling with preserving temporal dependence. Although their results for MSR-Action 3D are a little bit lower in terms of Chamfer Distance, they don't estimate temporal coherence, and probably the TPU-GAN is significantly better in this metric. Of course, the authors could find even more advantages.
2. Papers [1, 2, 3] consider Hausdorff Distance and Point-to-surface distance as metrics. Is it reasonable to include them in the analysis?
3. In equation (2), what is $\gamma$ and $h$? Are they similar to the same in the equation (1)?
4. In equation (3), why is it used 3 as the threshold?
5. Please, proofread the work. Examples:
Abstract. “In this work…” - should be separated by a comma;
Section 2. “Compared to PU-GAN…” - seems should be to PU-GCN or the reference on PU-GAN should be added;
Section 2. “...the change of points with respect to time are not taken into consideration” - use “is” instead of “are”;

References:
1. Li, Ruihui, et al. "Pu-gan: a point cloud upsampling adversarial network." Proceedings of the IEEE/CVF International Conference on Computer Vision. 2019.
2. Qian, Yue, et al. "Pugeo-net: A geometry-centric network for 3D point cloud upsampling." European Conference on Computer Vision. Springer, Cham, 2020.
3. Wang, Kaisiyuan, et al. "Sequential point cloud upsampling by exploiting multi-scale temporal dependency." IEEE Transactions on Circuits and Systems for Video Technology (2021).
4. Liu, Xingyu, Charles R. Qi, and Leonidas J. Guibas. "Flownet3d: Learning scene flow in 3d point clouds." Proceedings of the IEEE/CVF Conference on Computer Vision and Pattern Recognition. 2019.
5. Prantl L. et al. Tranquil clouds: Neural networks for learning temporally coherent features in point clouds //arXiv preprint arXiv:1907.05279. – 2019.


**Summary Of The Paper:**

In the paper, the authors firstly introduced TPU-GAN, a GAN that provides a temporal coherence for point cloud sequence.  Another contribution is an upsampling module built into TPU-GAN. In contrast to other methods that produce almost uniform sampling, this probability module provides a more reasonable distribution of upsampling points.
The authors investigate proposed methods both quantitatively and qualitatively on two different datasets. They show that TPU-GAN performs better than other state-of-the-art methods in terms of static view (per-frame) and temporal coherence. They also prove that the upsampling model in TPU-GAN provides distribution closer to ground truth distribution.


**Summary Of The Review:**

The proposed methods are well-investigated from different aspects in two different datasets. The paper is well-structured and well-written; most of the claims and hypotheses are supported by good arguments. However, it remains the question about the novelty's significance. If the authors provide reasonable explanations, I am ready to recommend the paper for acceptance.

Please, let me add a comment that I’m not an expert in the field of models for point clouds and admit that I could understand something wrong, but I've spent a lot of time reviewing the literature.

---

> ### Author Response · Authors · 2021-11-16
> **Response to reviewer 7Wbg**
>
> We would like to thank the reviewer for the efforts and constructive feedback. Below we would like to address the concerns.
>
> ### Major comments:
>
> * Q: **The background behind the TPU-GAN is based on existing approaches.**
>
> We would like to emphasize our main contribution in this work lies in the following aspects: a) To the best of our knowledge, we propose the first general spatial-temporal GAN framework for 3D point cloud sequences from different data sources; b) We propose an efficient adaptive upsampling strategy for non-uniform point clouds.
>
> Moreover, despite our work and Tranquil Clouds both use triplet of frames during training, our training setting is very different from Tranquil Clouds. We do not require the scene flow annotation (points’ displacement) during training. In addition, our method can use arbitrary frames for training (e.g. changing the number of input frames to 2, 4, 5, etc), but we found using 3 frames provides the best performance on two datasets.
>
> * Q: **Clarification on the difference between our method and Tranquil Clouds.**
>
> To clarify, the point correspondence across frames refers to, knowing what a point at the current frame will be located in the next frame. For instance, given two point clouds (each with two points) from consecutive frames: the first frame: [A, B], the second frame: [C, D], point correspondence across frames means that we know A would appear as D in the second frame (and B would appear as C). This information is needed in Tranquil clouds to evaluate point displacement across frames (first-order finite difference) and point acceleration across frames (second-order finite difference), but not necessary for our method. In practice, this point correspondence information is very difficult to obtain.
>
> ### Minor comments:
>
> * Q: **Missing related work: "Sequential point cloud upsampling by exploiting multi-scale temporal dependency".**
>
> We appreciate the reviewer for pointing out this relevant work, we have added the discussion about the above work in our revised manuscript.
>
> Sequential point cloud upsampling by exploiting multi-scale temporal dependency (SPU) presents a recurrent alignment module for assimilating information across frames into the point cloud upsampling process, which can be incorporated with existing point cloud upsampling networks. Both their work and our work leverage information across frames to augment the single-frame upsampling process. The main differences are that: a) their work focus on the generator side (augment the input feature with temporal information extracted by recurrent network), while our work focus on the discriminator side (use discriminator to guide generator); b) SPU assumes uniform upsampling as they use uniformity loss during training, whereas our work upsample points adaptively;
>
> As for the performance gap, we hypothesize the main reason is that in their work they set the upsampling ratio to be 4 on MSR-Action3D (256 input points), while in our work we set the upsampling ratio to be 16 (128 input points).
>
> * Q: **Papers [1, 2, 3] consider Hausdorff Distance and Point-to-surface distance as metrics. Is it reasonable to include them in the analysis?**
>
> Thanks for the suggestion. We have added the evaluation of Hausdorff distance in the revised manuscript (Table 1, 2, 4). For point-to-surface, as datasets we are using here do not have ground truth surface meshes, thus it is not applicable.
>
> * Q: **In equation (2), what is $\gamma$ and $h$? Are they similar to the same in the equation (1)?**
>
> Yes, they are similar. In practice, they are all implemented as multi-layer perceptions.
>
> * Q: **In equation (3), why is it used 3 as the threshold?**
>
> Our observation is that if an input point can only find 0∼2 neighbors in the ground truth point cloud, then it is very unlikely we need to upsample this point. Since upsampling it into 8 or 16 points (depending on predefined ratio) will introduce redundant points and therefore violate the ground truth point distribution.
>
> * Q: **“In this work…” - should be separated by a comma;**
>
> **“Compared to PU-GAN…” - seems should be to PU-GCN or the reference on PU-GAN should be added;**
>
> **“...the change of points with respect to time are not taken into consideration” - use “is” instead of “are”;**
>
> Thanks for the catch. They have now been corrected in the revised manuscript.

---

> > ### Comment · Reviewer_7Wbg · 2021-11-22
> > **Discussion**
> >
> > I appreciated the authors' explanations and found clarification for the majority of my questions:
> > 1. Differences between TPU-GAN and other existed methods (Tranquil clouds, SPU);
> > 2. Hyperparameters choosing for upsampling layer;
> > 3. Involving Hausdorff distance in considered metrics;
> > 4. General paper improvement.
> >
> > I also see that the authors include several supplementary materials with different method's evaluation experiments:
> > 1. Experiments on a self-driving dataset;
> > 2. Visualization and study on masking module;
> > 3. Visualization of generated point clouds.
> >
> > To sum up, I suppose that paper develops forward methods for a case of temporal point clouds. The supplementary experiments improve it in the positive direction.

---

### Official Review · Reviewer_dDgo · 2021-11-02

**Correctness:** 3
**Technical Novelty And Significance:** 2
**Empirical Novelty And Significance:** 3
**Recommendation:** 6
**Confidence:** 3

**Main Review:**

Strengths:

1. The problem addressed by the paper is significant. Leveraging temporal cues for point cloud upsampling is interesting and meaningful.

2. The approach makes sense. Approaches that leverage scene flow is likely to add too much computational burden and the annotation is very expensive to obtain. Using a discriminator can be a useful alternative.

3. The tasks are very interesting and commercially valuable. I am especially fond of the fluid mechnics simulation, and the potential commercial value behind it is immense.

4. The paper is coherent and clear, it is easy to follow.

5. Effective results in the visualization (Figure 2 and 3) where the proposed method clearly has a competitive edge.

6. Suitable baselines are chosen in the experiments: PU-GCN (static SOTA) and Tranquil Clouds (first learning-based temporal method).

Weaknesses:

1. The method is largely a combination of existing works. The basic modules may take less space and have their own "preliminaries" section: set abstraction that is taken from PointNet++, multi-scale graph convolution from PU-GCN, and temporal feature extraction module flow embedding from Flownet3d. More importantly, the critical part of the nolvelty, the temporal and spatial discriminators have simplistic designs by putting the above-mentioned modules together. This is may be the first work to leverage temporal information with discriminators, having a straight-forward design is understandable, but adaptation to the given tasks may improve the relevance.

2. The experiment setting may raise some questions, and the quantitive evaluation seems to have some performance gap. I understand that existing datasets for point cloud upsampling are all static, but I would like to hear the authors' comments on if it makes sense to tweak the conventional setting and introduce motion into the existing datasets (e.g. sample point clouds with a moving virtual camera or sample point clouds from a moving object, as these settings have some use cases in real life). Moreover, the performance of the proposed methods are not outperforming existing methods entirely (Table 1 and 2).


Minor comments:

1. Please elaborate on how duplicate points are removed in Figure 1a)

2. Sec 3.3: "often no correspondences" -> "often no clear correspondences". As there are definately some correspondences but we are unable to know.

3. Sec 3.3: "... remains unknown" -> "are not annotated"

4. Sec 3.4: "However, not all point clouds are distributed in a uniform way such that every point has similar amount of points in their neighborhood": solid observation, but this is also the case for static point cloud upsampling. Does point cloud sequence makes this problem more prominent? I would assume adaptive upsampling is especially critical for point cloud sequences as it is highlighted in the abstract.

5. Sec 3.5: lambda 1 and lambda 2 are different for different datasets, does this mean the hyperparameters are sensitive?

6. Sec 4.1 Dataset: why do you use different upsampling ratio for two tasks?

7. Would authors release the dataset for fluid simulation?

8. Sec 4.2 Comparison with other models: it is claimed that TPU-GAN outperforms PU-GCN on "high-level" distance metrics, does this make CD not an effective metric?

9. Page 8 footnote 2: input size is a critical factor, would it be possible to adapt the code of Tranquil Clouds to allow for 16x upsampling?

10. Conclusion and Figure 4 captions are excessively close to each other.

**Summary Of The Paper:**

This paper proposes to use a discriminator to enhance the temporal coherence for point cloud upsampling. The technique is tested in two scenarios: fluid mechanics simulation and human scan.

**Summary Of The Review:**

The paper has clearly several weaknesses, but it is undeniable that it has brought forward an interesting topic of point cloud upsampling with temporal cues, and developed two relevant tasks that are useful in real life. I thus opt for "marginally above the acceptance threshold" as the initial rating and hope the authors could address my concerns during the rebuttal period.

---

> ### Author Response · Authors · 2021-11-16
> **Response to reviewer dDgo: 1**
>
> We would like to thank the reviewer for the detailed comments. Here we address concerns as below.
>
> ### Weaknesses:
>
> * Q: **The novelty of the proposed method.**
>
> Our main contribution in this work includes: a) To the best of our knowledge, we propose the first spatial-temporal GAN framework for 3D point cloud sequences; b) We propose an efficient adaptive upsampling strategy for non-uniform point clouds.
>
> The simplicity of the model enables us to apply the proposed method to very different datasets (fluids, human action and Kitti scene flow in the revised manuscript) without heavy tweaking. On dense point cloud datasets like fluids (10000 input points and 80000 output points), the training of the model can be handled with moderate computing power (single GTX-1080Ti for 40 hours).
>
> As suggested there is indeed a lot of direction to extend and tweak the model for specific datasets. For instance, adding physic informed loss for fluid dataset, or using recurrent architecture in the generator when dealing with point clouds that are not so large. But here our focus is to build a general framework that could be generalized to various point cloud sequences.
>
> * Q: **Tweak existing static dataset.**
>
> It is very reasonable. There are already some existing datasets that make static objects into moving ones by applying simple rigid motion to them. For instance, moving MNIST and FlyingThings3D. Yet the motion in these datasets is relatively simple as the objects seldom deform (or deform only in a small scale), and motions are dominated by translations. It would be nteresting to extend these datasets to deformable and non-uniform point clouds such as fluids we have investigated in the paper, or other more complex scenarios like multiple interacting deformable objects.
>
> Sampling point clouds from objects under a moving camera is another very interesting and promising direction as it has a lot of potential practical applications. However when the camera is moving, at each snapshot the shape of the object is very likely to be incomplete and thus require additional processing techniques like point cloud completion, which is beyond the current scope of our proposed method.
>
> * Q: **The proposed method does not entirely outperform other methods.**
>
> In general, our proposed method is outperformed by PU-GCN on both datasets in terms of Chamfer distance. The main reason is that PU-GCN is solely trained on Chamfer distance loss and did not take other losses or temporal consistency into account. The training of TPU-GAN balances Chamfer distance with spatial and temporal discriminator losses. In addition, lower Chamfer distance does not always indicate better performance (Please see our comment on the question "*does this make CD not an effective metric?*" for more details).
>
>
> ### Minor comments:
>
> * Q: **Please elaborate on how duplicate points are removed in Figure 1a)**
>
> The upsampling module will predict an upsampling residual with shape: [N, 3r] (N: number of points, r: upsampling ratio, 3: dimension), and the masking module will predict a mask with shape: [N, 1]. If the mask value for a point is 0, then for its corresponding residual we only kept the first 3  and discard the rest 3r-3.
>
> * Q: **"often no correspondences" -> "often no clear correspondences". As there are definitely some correspondences but we are unable to know.**
>
> **"... remains unknown" -> "are not annotated"**
>
> Thanks for the suggestions. These sentences have now been adjusted.
>
> * Q: **Does point cloud sequence make this problem (non-uniform distribution) more prominent?**
>
> In general, non-uniform point clouds are very common whether they are sequential or not. Yet under the context of static point cloud upsampling, most currently available datasets are created from objects’ meshes by doing uniform Poisson disk sampling, which always creates a uniform point cloud. In addition, for many sequential point cloud data, like fluids, the surface is deforming violently and density changes every step, exploring a way to capture this irregular density distribution is of great importance to both visual plausibility and physical correctness.
>
> * Q: **Sec 3.5: lambda 1 and lambda 2 are different for different datasets, does this mean the hyperparameters are sensitive?**
>
> They are different across datasets mainly because the point distances in different datasets are very different (e.g. Chamfer distance for fluids is much smaller than scanned action data), and this results in different scales of Chamfer distance and masking loss. In practice, we just chose these hyperparameters such that the magnitudes of corresponding losses fall into the interval of [0, 1].
>
> **(Continued next reply due to character limit)**

---

> > ### Author Response · Authors · 2021-11-16
> > **Response to reviewer dDgo: 2**
> >
> > (Following the last reply)
> >
> > * Q: **why do you use different upsampling ratio for two tasks?**
> >
> > The main reason why we chose 8 as the upsampling ratio for the fluid dataset is that, we can more easily find the coarse-grained/fine-grained counterpart to compare with, since this ratio is the cube of 2 (2x2x2). For instance, given a fluid simulation with particle radius equalling 0.0250, its fine-grained counterpart should have a particle radius of 0.0125.
> >
> > For 16, this is roughly the largest ratio most point cloud upsampling works have adopted.
> >
> > * Q: **Would authors release the dataset for fluid simulation?**
> >
> > We will release the source code of the project later on. For now, the fluid dataset  (in .npz format) can be accessed via the following anonymized link:
> >
> > > Fluid dataset: https://drive.google.com/file/d/1RONVSpn978YpxZMVbIXt6JdVNpYG6c42/view?usp=sharing
> >
> > > Bunny/dragon scene:
> > https://drive.google.com/file/d/1uOtgekOsf9NxHswq73PRV_DpAejR69fb/view?usp=sharing
> >
> >
> > * Q: **It is claimed that TPU-GAN outperforms PU-GCN on "high-level" distance metrics, does this make CD not an effective metric?**
> >
> > We would like to offer our comment on this question from two perspectives.
> >
> > First of all, the downside of Chamfer distance (CD) has been investigated in paper [1], it is relatively biased compared to Earth Mover distance (EMD) when evaluating the similarity of dense point clouds. In this case, lower CD does not always indicate better performance.
> >
> > On the other hand, CD is still a practically very useful metric, as it is very straightforward to evaluate and can be evaluated exactly. This allows CD to be adopted as an exact metric for different datasets across different papers. As a comparison, EMD can only be evaluated based on approximation algorithm (auction-based algorithm, Sinkhorn) and is very sensitive to predefined hyperparameters like maximum iterations or convergence threshold. These parameters are usually sensitive to different data sources and do not guarantee the algorithm to converge [2].
> >
> > [1] Minghua Liu et al., Morphing and Sampling Network for Dense Point Cloud Completion, AAAI 2020
> >
> > [2] Earth Mover's Distance (EMD) loss #211, https://github.com/facebookresearch/pytorch3d/issues/211
> >
> > * Q: **Would it be possible to adapt the code of Tranquil Clouds to allow for 16x upsampling?**
> >
> > Thanks for pointing this out. We have now provided the results for both the 8x and 16x upsampling Tranquil clouds in the revised version of the paper.
> >
> > * Q: **Conclusion and Figure 4 captions are excessively close to each other.**
> >
> > Thanks for the suggestion. We have adjusted the margin between them in our revised manuscript.

---

> > > ### Comment · Reviewer_dDgo · 2021-11-21
> > > **Thank you for the discussion**
> > >
> > > I would like to thank the authors for the fruitful discussion. I feel most of my concerns have been addressed:
> > > 1. Experiment settings (dataset choice with added scene flow, lambdas, 16x Transquil Clouds)
> > > 2. Evaluation methods (by the way, I feel training on CD makes the method better at CD is a strong enough explanation as it is also observed training on EMD makes the model better at EMD for point cloud completion.)
> > >
> > > Some of the concerns are not fully addressed but I think they are not sufficient to undermine the contributions of the paper.
> > > 1. The simplicity of the methods. Although the methods are not impressive, they serve the purpose as the first approach towards point cloud sequence upsampling for general tasks.
> > > 2. The performance. I believe more tolerance should be given to new ideas that may not be that strong at the moment. These works may inspire many more others along the same direction.

---

### Official Review · Reviewer_waNz · 2021-11-03

**Correctness:** 3
**Technical Novelty And Significance:** 3
**Empirical Novelty And Significance:** 3
**Recommendation:** 6
**Confidence:** 4

**Main Review:**

Overall, the main issues of this paper are missing hyper-parameters experiments and unsatisfactory presentation quality. The authors are suggested to provide fair comparison under reasonable experimental settings, and improve the readability of the paper and further investigate the hyper-parameters to verify the robustness of the proposed method.


-Methods
- In Sec.3.1, the author mentioned only Inception DenseGCN is applied to the generator for efficiency. I am wondering how is the performance using the entire PU-GCN in the feature extractor? albeit the computational cost would increase.
- In Sec.3.2, it is not clear that how the author obtain the $r_{j}^{t+1}$  according to $r_{i}^{t}$. Do you require mapping points in the current frame into the corresponding points in the next frame, or using the same position in two frames directly?
- In Sec. 3.4, two parameters are introduced to the training process,(the predefined threshold $\epsilon$ in equal.3 and the number ’3’ in equal.4 ), please justify your hyperparameter choice in the paper.
- In Sec.3.5, the CD loss is used to evaluate the distance between generator output and ground truth due to its efficiency. How about using other losses? Could it bring huge computational costs?
- In Sec.3.5, equal.6 and equal.12 have the same symbol $L_G$, which may cause confusion, It seems an adversarial loss was missed in your paper, please clarify.


Experiments
1. Missing ablation:
- Explore up-sampling strategy: For up-sampling points, equal.5 requires the points to satisfy two conditions simultaneously. How about using equal.3 or equal.4 for up-sampling solely?
- Each component of equal.13 shall be studied to verify the effectiveness of each block.
2. Add visualization results: It is interesting that this work adopts a GAN to achieve super-resolution of points cloud. For this block, It would be better and convincing if the input and output point cloud during the training stage (for example, the first epoch and the final epoch) could be provided.


**Summary Of The Paper:**

This paper proposes a framework called Temporal Point cloud Up-sampling to capture temporal and spatial features, in light of the difficulty to obtain point correspondence annotation. The framework is constituted by a generator for refining coarse point cloud, a temporal, and a spatial discriminator for feature extracting. Experiments conducted on two datasets towards human action recognition and fluid particles analysis demonstrate the effectiveness of this method.

**Summary Of The Review:**

As mentioned in the main review, I am inclined to weakly reject this paper due to the missing hyper-parameters experiments, unsatisfactory presentation quality.

---

> ### Author Response · Authors · 2021-11-16
> **Response to reviewer waNz**
>
> We would like to thank the reviewer for the constructive feedback and here we address concerns as below.
>
> ### Methods:
> * Q: **How about using the entire PU-GCN in the feature extractor?**
>
> We think the statement in Sec.3.1 might be unclear and has caused confusion here. We have improved the illustration in our revised manuscript. To clarify, PU-GCN uses multiple layers of Inception DenseGCN as its feature extractor, but PU-GCN itself is not a feature extraction unit as it comprises a feature extractor and also a coordinate reconstructor that predicts upsampled points. We use Inception DenseGCN layer in the feature extractor of our generator but not in the discriminators due to its computational cost.
>
> * Q: **It is not clear that how the author obtain $r_j^{t+1}$ the according to $r_i^{t}$.**
>
> Given two point clouds from two consecutive frames $\{P^t, P^{t+1}\}$, to obtain  $r_j^{t+1}, j \in P^{t+1}$, we place $r_i^{t},  i \in P^{t}$ into the points of the next frame $P^{t+1}$ and do neighbor searching (either via k-nearest neighbor or fixed radius neighbor).
>
> * Q: **Please justify your hyperparameters.**
>
> Thanks for the suggestion. We have added explanations of these hyperparameter choices in our revised manuscript.
>
>  The first hyperparameter **$\eta$** is chosen heuristically and actually, it can be set very loosely (e.g. 0.05, 0.1, 0.5) without influencing final results. In practice, we observe over 99% of points have mask value either above 0.99 or below 0.01.
>
> The second hyperparameter **“3”** is chosen based on the following observation:
> Given a sparse point cloud $P$ and predefined upsampling ratio $r$, every point $i$ in $P$ will be upsampled into $r$ points and these $r$ points will be very close to their original input point $i$ (because of the locality of upsampling network). For non-uniform point clouds, it would be unnecessary to upsample points that could not find sufficient neighbors in the ground truth, as upsampling them into 8 or 16 points (depending on the predefined ratio) would introduce redundant points. Therefore we mark those points which can only find 0~2 neighbors in the ground truth as “no need to upsample”, and set the threshold as 3.
>
> * Q: **Compared to CD, how about other losses? Could it bring huge computational costs?**
>
> Yes, compared to another widely used distance metric Earth Mover Distance (EMD), CD is much faster. EMD is very expensive when used to train model on dense point clouds, and therefore in the paper we only use it for offline evaluation.
>
> Below we provide a benchmark of these two loss functions.
>
> Point cloud size: [8, 80000, 3] (batch size, num of points, dimension), 1000 evaluations on a 1080 Ti GPU, EMD is evaluated using the auction-based algorithm with a maximum iteration of 100.
>
> | EMD: **0.9381** sec per evaluation |    CD:**0.0003** sec per evaluation |
>
> * Q: **In Sec.3.5, equal.6 and equal.12 have the same symbol, which may cause confusion and seems an adversarial loss is missed.**
>
> Thanks for pointing it out. We have removed equal.6 to avoid causing confusion in our revised manuscript.
>
> To clarify, the adversarial loss for temporal discriminator is defined in equal.6 (equal. 8 in original version), for spatial discriminator is defined in equal.7 (equal. 9 in original version). The generator part is defined in equal.8&9 (equal. 10&11 in original version).
>
> ### Experiments
> * Q: **How about using equal.3 or equal.4 for up-sampling solely?**
>
> equal.5 is used to create the labels for directly supervising the training of masking module.  Without this masking loss (equal.5), the weights of the masking module can only receive gradients from CD loss and adversarial losses. In practice, we find this will cause the masking module to learn a trivial solution, that is to predict a mask with many values very close to zero and make points clustered togther. (Please refer to Fig.9 in section A.6 of manuscript for more details)
>
> * Q: **Each component of equal.13 shall be studied to verify the effectiveness of each block.**
>
> The ablation of different modules is originally presented in Sec.4.4 in the paper, we are happy to address or improve this part if reviewer has further suggestions.
>
> * Q: **Add visualization results during training stage.**
>
> We appreciate reviewer’s suggestion on adding visualizations. We have now added visualizations of point clouds generated during the training process. (Please refer to Fig.11, Fig.12 in section A.8 of the manuscript).

---

### Author Response · Authors · 2021-11-16
**General response to all reviewers**

We want to thank all the reviewers for their insightful comments on our work.

Here we would like to clarify our main contribution:
1. **Simplicity**:
Our primary focus is to develop a generalized upsampling framework for point cloud sequences. Despite its simplicity, we demonstrate the effectiveness of our proposed framework on very different data sources -fluids, human action (and Kitti scene flow added in the revised manuscript) without much tweaking.

2. **Dual discriminator for point cloud sequence**:
Despite dual discriminator setting has become a common technique in video modeling, to the best of our knowledge, we are the first work that introduces dual discriminator into generative modeling of point cloud sequences.

3. **Adaptive upsampling**:
Many existing learning methods assume upsampled points should be uniformly distributed. In this work, we propose a simple but efficient method to deal with irregular point clouds that has non-uniform distribution.


We have made an update to the paper including the following changes:

#### Major changes:
* Added visualization of point cloud generated during the training process. (Section A.8 in the appendix)
* Added visualization on the masking module. (Section A.6 in the appendix)
* Added an experiment on the self-driving dataset (Kitti scene flow) and compared with other baselines. We show that our model can also benefit downstream tasks like scene flow prediction when data is limited and sparse. (Section A.9 in the appendix)

#### Minor changes:
* Fixed typos
* Sentence adjustment
* Added Hausdorff distance as an additional metric  (Table 1, 2, 4).
* Added missing related works
* Added evaluation result for 16x upsampling Tranquil Clouds

We have also addressed detailed comments of every reviewer in separate replies. Please don’t hesitate to let us know if there is any further question or additional comment.

---

### Decision · Program_Chairs · 2022-01-20

**Decision:**

Accept (Poster)

**Comment:**

The paper proposes a GAN framework for dynamic point cloud superresolution. It does not need scene flow supervision for training and has an interesting adaptive upsampling mechanism. Results are shown on several datasets and are reasonably convincing. Overall, all the reviewers are slightly positive about the work. After the rebuttal, all the five reviewers converged to a marginally-above-the-threshold recommendation. The meta-reviewer agreed with their assessment and would like to recommend accepting the paper.